# Glycan processing in the Golgi as optimal information coding that constrains cisternal number and enzyme specificity

**Alkesh Yadav[1], Quentin Vagne[2], Pierre Sens[2], Garud Iyengar[3]\*, Madan Rao[4]\***

[1]Raman Research Institute, Bangalore, India; [2]Laboratoire Physico Chimie Curie, Institut Curie, CNRS UMR168, Paris, France; [3]Industrial Engineering and Operations Research, Columbia University, New York, United States; [4]Simons Centre for the Study of Living Machines, National Centre for Biological Sciences, Bangalore, India

**Abstract** Many proteins that undergo sequential enzymatic modification in the Golgi cisternae are displayed at the plasma membrane as cell identity markers. The modified proteins, called glycans, represent a molecular code. The fidelity of this *glycan code* is measured by how accurately the glycan synthesis machinery realizes the desired target glycan distribution for a particular cell type and niche. In this article, we construct a simplified chemical synthesis model to quantitatively analyse the trade-offs between the number of cisternae, and the number and specificity of enzymes, required to synthesize a prescribed target glycan distribution of a certain complexity to within a given fidelity. We find that to synthesize complex distributions, such as those observed in real cells, one needs to have multiple cisternae and precise enzyme partitioning in the Golgi. Additionally, for a fixed number of enzymes and cisternae, there is an optimal level of specificity (promiscuity) of enzymes that achieves the target distribution with high fidelity. The geometry of the *fidelity landscape* in the multidimensional space of the number and specificity of enzymes, inter-cisternal transfer rates, and number of cisternae provides a measure for robustness and identifies stiff and sloppy directions. Our results show how the complexity of the target glycan distribution and number of glycosylation enzymes places functional constraints on the Golgi cisternal number and enzyme specificity.

**\*For correspondence:**
garud@ieor.columbia.edu (GI);
madan@ncbs.res.in (MR)

## Editor's evaluation

This article contributes to an important and largely unexplored topic in cell biology: the understanding of glycosylation. The authors introduce a mathematical model of glycosylation in the Golgi apparatus and use the model to investigate how the complexity (diversity) and fidelity of the plasma membrane glycan distribution depend on parameters such as the number of Golgi cisternae or enzyme specificity. The article is well written and makes the effort to present a rather complex topic in an accessible way by leaving some of the details in the appendices.

## Introduction

A majority of the proteins synthesized in the endoplasmic reticulum (ER) are transferred to the Golgi cisternae for further chemical modification by glycosylation (**Alberts, 2002**), a process that sequentially and covalently attaches sugar moieties to proteins, catalyzed by a set of enzymatic reactions within the ER and the Golgi cisternae. These enzymes, called glycosyltransferases, are localized in the ER and *cis*-medial and *trans*-Golgi cisternae in a specific manner (**Varki, 2009**; **Cummings and Pierce, 2014**). Glycans, the final products of this glycosylation assembly line, are delivered to the

plasma membrane (PM) conjugated with proteins, whereupon they engage in multiple cellular functions, including immune recognition, cell identity markers, cell-cell adhesion, and cell signalling (*Varki, 2009*; *Cummings and Pierce, 2014*; *Varki, 2017*; *Drickamer and Taylor, 1998*; *Gagneux and Varki, 1999*). This *glycan code* (*Gabius, 2018*; *Dwek, 1996*), representing information (*Winterburn and Phelps, 1972*) about the cell, is generated dynamically, following the biochemistry of sequential enzymatic reactions and the biophysics of secretory transport (*Varki, 2017*; *Varki, 1998*; *Pothukuchi et al., 2019*).

In this article, we will focus on the role of glycans as markers of cell identity. For the glycans to play this role, they must inevitably represent a molecular code (*Gabius, 2018*; *Varki, 2017*; *Pothukuchi et al., 2019*). While the functional consequences of glycan alterations have been well studied, the glycan code has remained an enigma (*Gabius, 2018*; *Pothukuchi et al., 2019*; *Bard and Chia, 2016*; *D'Angelo et al., 2013*). We study the *fidelity* of molecular code generation, that is, the precision and reliability with which the glycan distribution is created. While it has been recognized that fidelity of the glycan code is necessary for reliable cellular recognition (*Demetriou et al., 2001*), a quantitative measure of fidelity of the mechanism and the constraints that fidelity requirements put on cellular structure and organization are lacking.

There are two aspects of the cell-type-specific glycan code and the code generation mechanism that have an important bearing on quantifying fidelity. The first is that extant glycan distributions have high *complexity* (section 'Complexity of glycan code'), owing to evolutionary pressures arising from (a) reliable cell-type identification amongst a large set of different cell types in a complex organism, the preservation and diversification of 'self-recognition' (*Drickamer and Taylor, 1998*), (b) pathogen-mediated selection pressures (*Varki, 2009*; *Varki, 2017*; *Gagneux and Varki, 1999*), and (c) *herd immunity* within a heterogenous population of cells of a community (*Wills and Green, 1995*) or within a single organism (*Drickamer and Taylor, 1998*). We interpret this to mean that the *target distribution* of glycans for a given cell type is complex; in section 'Complexity of glycan code', we define a quantitative measure for complexity and demonstrate its implications. The second is that the cellular machinery for the synthesis of glycans, which involves sequential chemical processing via cisternal resident enzymes and cisternal transport, is subject to variation and noise (*Varki, 2017*; *Varki, 1998*; *Pothukuchi et al., 2019*); the *synthesized glycan distribution* is, therefore, a function of cellular parameters such as the number and specificity of enzymes, inter-cisternal transfer rates, and number of cisternae. We will discuss an explicit model of the cellular synthesis machinery in section 'Synthesis of glycans in the Golgi cisternae'.

Here, we define *fidelity* as the minimum achievable Kullback-Leibler (KL) divergence (*Cover and Thomas, 2012*; *MacKay, 2003*) between the synthesized distribution of glycans and the target glycan distribution as a function of given cellular parameters, such as the number and specificity of enzymes, inter-cisternal transfer rates, and number of cisternae (section 'Optimization problem'). Using a simplified chemical synthesis model, we analyse the trade-offs between the number of cisternae and the number and specificity of enzymes in order to achieve a prescribed target glycan distribution with high fidelity (section 'Results of optimization'). Our analysis leads to a number of interesting results, a few of which we list here:

i. First, since an important function of the glycan spectrum is cell type/niche identification, it seems natural to relate *glycan complexity* to organismal complexity taken to be associated with the number of cell types in the organism (*Carroll, 2001*; *Bonner, 1998*). Here, we provide a *measure of the complexity* of the glycan distribution of a given cell type using mass spectrometry coupled with determination of molecular structure (MSMS) data. Using this we have analysed the MSMS data from hydra, planaria, and mammalian cells. We find that the complexity of the glycan distribution indeed correlates with the organism complexity.

ii. Constructing a high-fidelity representation of a *complex target distribution*, such as those observed in real cells, requires a *complex Golgi machinery* with multiple cisternae, precise enzyme partitioning, and control on enzyme specificity. This definition of fidelity of the glycan code allows us to provide a quantitative argument for the evolutionary requirement of multiple compartments. While it is possible to produce complex glycan distributions in one compartment using a large number of enzymes, such a design would inevitably require a more elaborate genetic cost.

iii. Within our synthesis model, an increase in the number of Golgi cisternae drives an increase in the glycan complexity, keeping everything else fixed.

iv. We explore the geometry of the *fidelity landscape* in the multidimensional space of the number and specificity of enzymes, inter-cisternal transfer rates, and number of cisternae. This allows us to discuss issues such as *robustness* to noise, and *stiff and sloppy directions* in this multidimensional space.

v. For fixed number of enzymes and cisternae, there is an optimal level of specificity of enzymes that achieves the complex target distribution with high fidelity. Keeping the number of enzymes fixed, having low specificity or sloppy enzymes and larger cisternal number could give rise to a diverse repertoire of functional glycans, a strategy used in organisms such as plants and algae. Promiscuous enzymes bring in the potential for *evolvability* (***Kirschner and Gerhart, 2008***); promiscuity allows the system to be stable to random mutations in proteins or variations in the target distribution.

Thus, our results imply that the pressure to produce the target glycan code for a given cell type with high fidelity places strong constraints on the cisternal number and enzyme specificity (***Sengupta and Linstedt, 2011***). Taken together, our quantitative analysis of the trade-offs has deep implications for the non-equilibrium self-assembly of Golgi cisternae and suggests that the control of cisternal number must involve a coupling of non-equilibrium self-assembly of cisternae with enzymatic chemical reaction kinetics (***Glick and Malhotra, 1998***). This combined dynamics of chemical processing with non-equilibrium membrane dynamics involving fission, fusion, and transport (***Sachdeva et al., 2016***; ***Sens and Rao, 2013***) opens up a new direction for future research.

## Complexity of glycan code

Since each cell type (in a niche) is identified with a distinct glycan profile (***Gabius, 2018***; ***Varki, 2017***; ***Pothukuchi et al., 2019***), and this glycan profile is noisy because of the stochastic noise associated with the synthesis and transport (***Pothukuchi et al., 2019***; ***Bard and Chia, 2016***; ***D'Angelo et al., 2013***), a large number of different cell types can be differentiated only if the cells are able to produce a large set of glycan profiles that are distinguishable in the presence of this noise. Our task is to identify a quantitative measure for the *complexity* of a glycan profile such that a set of more complex glycan profiles is able to support a larger number of well-separated profiles, and therefore, a larger number of cell types, or equivalently, a more *complex* organism (a rigorous definition of complexity can be given in terms of the KL metric [***Cover and Thomas, 2012***; ***MacKay, 2003***] between two

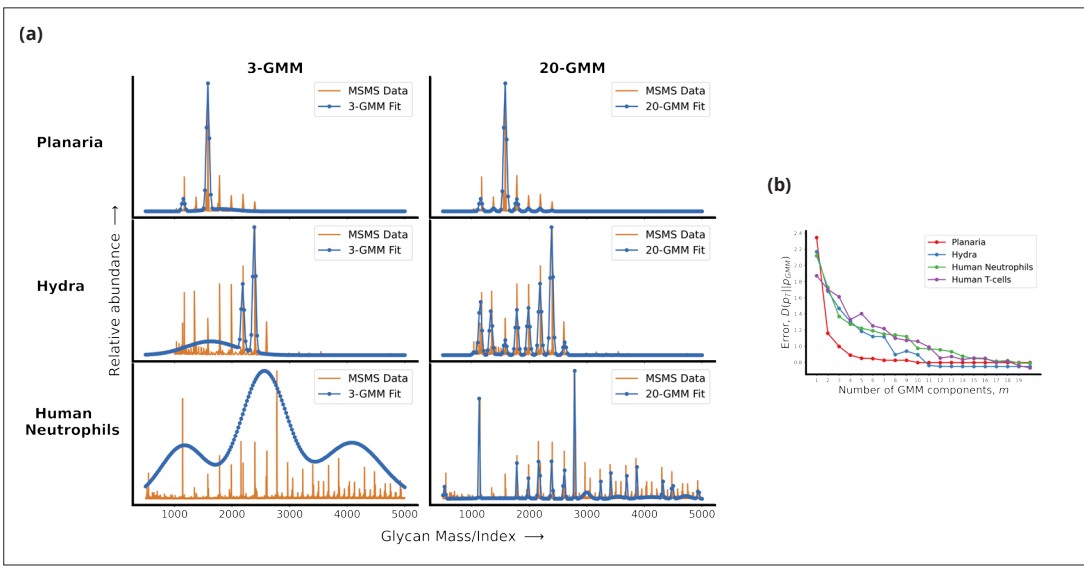

**Figure 1.** Living cells display a complex glycan distribution. (**a**) 3-Gaussian mixture model (GMM) and 20-GMM approximation for the relative abundance of glycans taken from mass spectrometry coupled with determination of molecular structure (MSMS) data of planaria *Schmidtea mediterranea*, *Hydra magnipapillata*, and *human* neutrophils. (**b**) The change in the Kullback–Leibler (KL) divergence $D(p_T \| p_{GMM}^{(m)})$ as a function of the number of GMM components $m$. The KL divergence for planaria saturates at $m = 5$, for hydra at $m = 11$, and for *human* cells at $m = 20$. Thus, the number of components required to approximate the glycan profile correlates well with the complexity of the organism. Details are given in Appendix 1.

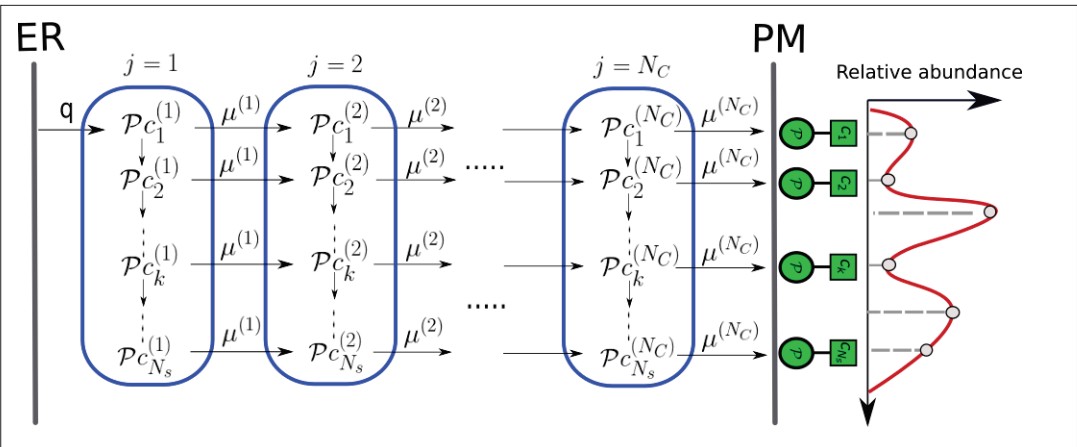

**Figure 2.** Enzymatic reaction and transport network in the secretory pathway. Represented here is the array of Golgi cisternae (blue) indexed by $j = 1, \ldots, N_C$ situated between the endoplasmic reticulum (ER) and plasma membrane (PM). Glycan-binding proteins $\mathcal{P}c_1^{(1)}$ are injected from the ER to cisterna-1 at rate $q$. Superimposed on the Golgi cisternae is the transition network of chemical reactions (column) – inter-cisternal transfer (rows), the latter with rates $\mu^{(j)}$. $\mathcal{P}c_1^{(1)}$ denotes the acceptor substrate in compartment $j$ and the glycosyl donor $c_0$ is chemostated in each cisterna. This results in a distribution (relative abundance) of glycans displayed at the PM (red curve), which is representative of the cell type.

glycan profiles. We declare that two profiles are distinguishable only if the KL distance between the profiles is more than a given tolerance. This tolerance is an increasing function of the noise. We define the *complexity* of a set of possible glycan profiles as the size of the largest subset such that the KL distance of any pair of profiles is larger than the tolerance). Furthermore, we would like to be able to estimate the complexity of a glycan profile from molecular structure (MSMS) measurements (*Cummings and Crocker, 2020*; *Subramanian et al., 2018*; *Sahadevan et al., 2014*).

In order to identify such a quantitative measure of complexity, we first need a consistent way of smoothening or coarse-graining the raw glycan profiles obtained from MSMS measurements to remove measurement and synthesis noise. Here, we denoise the glycan profile by approximating it by a Gaussian mixture model (GMM) with a specified number of components that are supported on a finite set of indices (*Bacharoglou, 2010*). Since the size of the set of all possible *m*-component Gaussian densities is an increasing function of *m*, we define the complexity of a mixture of Gaussians as the number of components *m*. *Figure 1* demonstrates that the value of *m* at which the *m*-component GMM approximation of the target profile saturates is a good measure of complexity. Using this definition we see that the complexity of the glycan profiles of various organisms correlates well with the number of cell types in an organism (details of the procedure are given in Appendix 1). We will now describe a general model of the cellular machinery that is capable of synthesizing glycans of any complexity. We expect that cells need a more elaborate mechanism to produce profiles from a more complex set.

## Synthesis of glycans in the Golgi cisternae

The glycan display at the cell surface is a result of proteins that flux through and undergo sequential chemical modification in the secretory pathway, comprising an array of Golgi cisternae situated between the ER and PM, as depicted in *Figure 2*. Glycan-binding proteins (GBPs) are delivered from the ER to the first cisterna, whereupon they are processed by the resident enzymes in a sequence of steps that constitute the N-glycosylation process (*Varki, 2009*). A generic enzymatic reaction in the cisterna involves the catalysis of a group transfer reaction in which the monosaccharide moiety of a simple sugar donor substrate, for example, UDP-Gal, is transferred to the acceptor substrate, by a Michaelis–Menten (MM)-type reaction (*Varki, 2009*)

$$
\begin{aligned}
&\text{Acceptor + glycosyl donor + Enzyme} \\
&\underset{\omega_b}{\overset{\omega_f}{\rightleftharpoons}} \left[\text{Acceptor} \cdot \text{glycosyl donor} \cdot \text{Enzyme}\right] \xrightarrow{\omega_c} \text{glycosylated acceptor + nucleotide + Enzyme}
\end{aligned} \quad (1)
$$

From the first cisterna, the proteins with attached sugars are delivered to the second cisterna at a given inter-cisternal transfer rate, where further chemical processing catalyzed by the enzymes resident in the second cisterna occurs. This chemical processing and inter-cisternal transfer continue until the last cisterna, thereupon the fully processed glycans are displayed at the PM (*Varki, 2009*). The network of chemical processing and inter-cisternal transfer forms the basis of the physical model that we will describe next.

Any physical model of such a network of enzymatic reactions and cisternal transfer needs to be augmented by reaction and transfer rates and chemical abundances. To obtain the range of allowed values for the reaction rates and chemical abundances, we use the elaborate enzymatic reaction models, such as the KB2005 model (*Umaña and Bailey, 1997*; *Krambeck et al., 2009*; *Krambeck and Betenbaugh, 2005*) (with a network of 22,871 chemical reactions and 7565 oligosaccharide structures) that predict the N-glycan distribution based on the activities and levels of processing enzymes distributed in the Golgi cisternae of mammalian cells. For the allowed rates of cisternal transfer, we rely on the recent study by Ungar and coworkers (*Fisher et al., 2019*; *Fisher and Ungar, 2016*), whose study shows how the overall Golgi transit time and cisternal number can be tuned to engineer a homogeneous glycan distribution.

## Model
### Chemical reaction and transport network in cisternae

We consider an array of $N_C$ Golgi cisternae, labelled by $j = 1, \ldots, N_C$, between the ER and PM (*Figure 2*). GBPs, denoted as $\mathcal{P}c_1^{(1)}$, are delivered from the ER to cisterna-1 at an injection rate $q$. It is well established that the concentration of the glycosyl donor in the $j$th cisterna is chemostated (*Varki, 2009*; *Hirschberg et al., 1998*; *Caffaro and Hirschberg, 2006*; *Berninsone and Hirschberg, 2000*), thus in our model we hold its concentration $c_0^{(j)}$ constant in time for the $j$th cisterna. The acceptor $\mathcal{P}c_1^{(1)}$ reacts with $c_0^{(1)}$ to form the glycosylated acceptor $\mathcal{P}c_2^{(1)}$, following an MM reaction (1) catalyzed by the appropriate enzyme. The acceptor $\mathcal{P}c_2^{(1)}$ has the potential of being transformed into $\mathcal{P}c_3^{(1)}$, and so on, provided the requisite enzymes are present in that cisterna. This leads to the sequence of enzymatic reactions $\mathcal{P}c_1^{(1)} \rightarrow \mathcal{P}c_2^{(1)} \rightarrow \ldots \mathcal{P}c_k^{(1)} \rightarrow \ldots$, where $k$ enumerates the sequence of glycosylated acceptors using a consistent scheme (such as in *Umaña and Bailey, 1997*). The glycosylated GBPs are transported from cisterna-1 to cisterna-2 at an inter-cisternal transfer rate $\mu^{(1)}$, whereupon similar enzymatic reactions proceed. The processes of intra-cisternal chemical reactions and inter-cisternal transfer continue to the other cisternae and form a network as depicted in *Figure 2*. Although, in this article, we focus on a sequence of reactions that form a line graph, the methodology we propose extends to tree-like reaction sequences, and more generally to reaction sequences that form a directed acyclic graph.

Let $N_s - 1$ denote the maximum number of possible glycosylation reactions in each cisterna $j$, catalyzed by enzymes labelled as $E_\alpha^{(j)}$, with $\alpha = 1, \ldots, N_E$, where $N_E$ is the total number of enzyme species in each cisterna. Since many substrates can compete for the substrate-binding site on each enzyme, one expects in general that $N_s \gg N_E$. The configuration space of the network in *Figure 2* is $N_s \times N_C$. For the N-glycosylation pathway in a typical mammalian cell, $N_s = 2 \times 10^4$, $N_E = 10$–$20$, and $N_C = 4$–$8$ (*Umaña and Bailey, 1997*; *Krambeck and Betenbaugh, 2005*; *Krambeck et al., 2009*; *Fisher and Ungar, 2016*). We account for the fact that the enzymes have specific cisternal localization by setting their concentrations to zero in those cisternae where they are not present.

The action of enzyme $E_\alpha^{(j)}$ on the substrate $\mathcal{P}c_k^{(j)}$ in cisterna $j$ is given by

$$\mathcal{P}c_k^{(j)} + E_\alpha^{(j)} \underset{\omega_b(j,k,\alpha)}{\overset{\omega_f(j,k,\alpha)c_0^{(j)}}{\rightleftharpoons}} \left[ E_\alpha^{(j)} - \mathcal{P}c_k^{(j)} - c_0^{(j)} \right] \overset{\omega_c(j,k,\alpha)}{\longrightarrow} \mathcal{P}c_{k+1}^{(j)} + E_\alpha^{(j)} \tag{2}$$

where $k = 1, \ldots N_s - 1$. In general, the forward, backward, and catalytic rates $\omega_f$, $\omega_b$, and $\omega_c$, respectively, depend on the cisternal label $j$, the reaction label $k$, and the enzyme label $\alpha$, which parametrize the MM reactions (*Price and Stevens, 1999*). For instance, structural studies on glycosyltransferase-mediated synthesis of glycans (*Moremen and Haltiwanger, 2019*) would suggest that the forward rate $\omega_f$ depends on the binding energy of the enzyme $E_\alpha^{(j)}$ to acceptor substrate $\mathcal{P}c_k^{(j)}$ and a *physical variable* that characterizes the cisternae.

A potential candidate for such a cisternal variable is pH (**Kellokumpu, 2019**), whose value is maintained homeostatically in each cisterna (**Casey et al., 2010**); changes in pH can affect the shape of an enzyme (substrate) or its charge properties, and in general the reaction efficiency of an enzyme has a pH optimum (**Price and Stevens, 1999**). Another possible candidate for a cisternal variable is membrane bilayer thickness (**Dmitrieff et al., 2013**); indeed, both pH (**Llopis et al., 1998**) and membrane thickness are known to have a gradient across the Golgi cisternae. We take $\omega_f(j, k, \alpha) \propto P^{(j)}(k, \alpha)$, where $P^{(j)}(k, \alpha) \in (0, 1)$ is the binding probability of enzyme $E_\alpha^{(j)}$ with substrate $\mathcal{P}c_k^{(j)}$, and define the binding probability $P^{(j)}(k, \alpha)$ using a biophysical model, similar in spirit to the Monod-Wyman-Changeux model of enzyme kinetics (**Monod et al., 1965**; **Changeux and Edelstein, 2005**) that depends on enzyme-substrate-induced fit.

Let $\ell_\alpha^{(j)}$ and $\ell_k$ denote, respectively, the optimal 'shape' for enzyme $E_\alpha^{(j)}$ and the substrate $\mathcal{P}c_k^{(j)}$. We assume that the mismatch (or distortion) energy between the substrate $k$ and enzyme $E_\alpha^{(j)}$ is $\|\ell_k - \ell_\alpha^{(j)}\|$, with a binding probability given by

$$P^{(j)}(k, \alpha) = \exp\left(-\sigma_\alpha^{(j)}\|\ell_k - \ell_\alpha^{(j)}\|\right) \tag{3}$$

where $\| \cdot \|$ is a distance metric defined on the space of $\ell_\alpha^{(j)}$ (e.g. the square of the $\ell_2$-norm would be related to an elastic distortion model [**Savir and Tlusty, 2007**]) and the vector $\boldsymbol{\sigma} \equiv [\sigma_\alpha^{(j)}]$ parametrizes enzyme *specificity*. This distortion model captures the above idea that the reaction between the flexible enzyme and fixed substrate is facilitated by an induced fit. A large value of $\sigma_\alpha^{(j)}$ indicates a highly specific enzyme, a small value of $\sigma_\alpha^{(j)}$ indicates a promiscuous enzyme. It is recognized that the degree of enzyme specificity or sloppiness is an important determinant of glycan distribution (**Varki, 2009**; **Roseman, 2001**; **Hossler et al., 2007**; **Yang et al., 2018**).

Our synthesis model is mean field, in that we ignore stochasticity in glycan synthesis that may arise from low copy numbers of substrates and enzymes, multiple substrates competing for the same enzymes, and kinetics of inter-cisternal transfer (**Umaña and Bailey, 1997**; **Krambeck et al., 2009**; **Krambeck and Betenbaugh, 2005**). Then the usual MM steady-state conditions for (2), which assumes that the concentration of the intermediate enzyme-substrate complex does not change with time, imply that

$$\left[E_\alpha^{(j)} - \mathcal{P}c_k^{(j)} - c_0^{(j)}\right] = \frac{\omega_f(j, k, \alpha)\, c_0^{(j)}}{\omega_b(j, k, \alpha) + \omega_c(j, k, \alpha)} E_\alpha^{(j)} c_k^{(j)}.$$

where $c_k^{(j)}$ is the *concentration* of the acceptor substrate $\mathcal{P}c_k^{(j)}$ in compartment $j$.

Together with the constancy of the total enzyme concentration, $\left[E_\alpha^{(j)}\right]_{tot} = E_\alpha^{(j)} + \sum_{k=1}^{N_s}\left[E_\alpha^{(j)} - \mathcal{P}c_k^{(j)} - c_0^{(j)}\right]$, this immediately fixes the kinetics of product formation (not including inter-cisternal transport),

$$\frac{dc_{k+1}^{(j)}}{dt} = \sum_{\alpha=1}^{N_E} \frac{V(j, k, \alpha)P^{(j)}(k, \alpha)c_k^{(j)}}{M(j, k, \alpha)\left(1 + \sum_{k'=1}^{N_s} \frac{P^{(j)}(k', \alpha)c_{k'}^{(j)}}{M(j, k', \alpha)}\right)} \quad k = 1, \ldots, N_S; \ j = 1, \ldots, N_C \tag{4}$$

where

$$M(j, k, \alpha) = \frac{\omega_b(j, k, \alpha) + \omega_c(j, k, \alpha)}{\omega_f(j, k, \alpha)c_0^{(j)}}P^{(j)}(k, \alpha)$$

and

$$V(j, k, \alpha) = \omega_c(j, k, \alpha)\left[E_\alpha^{(j)}\right]_{tot}.$$

This reparametrization of the reaction rates $\omega_f, \omega_b, \omega_c$ in terms of $\mathbf{M}, \mathbf{V}$ is convenient since it relates to experimentally measurable parameters $V_{max}$ and MM constant $K_M$, for each $(j, k, \alpha)$, which can be easily read out (see Appendix 2). As is the usual case, the maximum velocity $V_{max}$ is not an

intrinsic property of the enzyme because it is dependent on the enzyme concentration $\left[E_\alpha^{(j)}\right]_{tot}$; while $K_M(j,k,\alpha) = M(j,k,\alpha)c_0^{(j)}/P^{(j)}(k,\alpha)$ is an intrinsic parameter of the enzyme and the enzyme-substrate interaction. The enzyme catalytic efficiency, the so-called "$k_{cat}/K_M$" $\propto P^{(j)}(k,\alpha)$, is high for *perfect* enzymes (*Bar-Even et al., 2015*) with minimum mismatch.

We now add to this chemical reaction kinetics the rates of injection ($q$) and inter-cisternal transport $\mu^{(j)}$ from the cisterna $j$ to $j+1$; in Appendix 3, we display the complete set of equations that describe the changes in the substrate concentrations $c_k^{(j)}$ with time. These kinetic equations automatically obey the conservation law for the protein concentration ($p$). At steady state, these kinetic equations lead to a set of nonlinear recursion *equations (15)-(16)* that are displayed in Appendix 3, which can be solved numerically to obtain the steady-state glycan concentrations, $\mathbf{c} \equiv c_k^{(j)}$, as a function of the independent vectors $\mathbf{M} \equiv [M(j,k,\alpha)]$, $\mathbf{V} \equiv [V(j,k,\alpha)]$, and $\mathbf{L} \equiv [P^{(j)}(k,\alpha)]$, the transport rates $\boldsymbol{\mu} \equiv [\mu^{(j)}]$ and specificity, $\boldsymbol{\sigma} \equiv [\sigma_\alpha^{(j)}]$.

## Optimization problem

Let $\boldsymbol{c}^*$ denote the 'target' concentration distribution, normalized to the distribution so that $\sum_{k=1}^{N_s} c_k^* = 1$, for a particular cell type, that is, the goal of the sequential synthesis mechanism described in the section 'Chemical reaction and transport network in cisternae' is to approximate $\boldsymbol{c}^*$. Let $\bar{\boldsymbol{c}}$ denote the normalized steady-state glycan concentration distribution displayed on the PM. Then *Equation 16* implies that $\bar{c}_k = \mu^{(N_C)}c_k^{(N_C)}$, $k = 1, \ldots, N_s$. We measure the *fidelity* $F(\boldsymbol{c}^*\|\bar{\boldsymbol{c}})$ between the $\boldsymbol{c}^*$ and $\bar{\boldsymbol{c}}$ by the ratio of the KL divergence $D(\boldsymbol{c}^*\|\boldsymbol{c})$ (*Cover and Thomas, 2012*; *MacKay, 2003*) to the entropy $H(\boldsymbol{c}^*)$

$$F(\boldsymbol{c}^*\|\bar{\boldsymbol{c}}) := \frac{D(\boldsymbol{c}^*\|\bar{\boldsymbol{c}})}{H(\boldsymbol{c}^*)} = \frac{\sum_{k=1}^{N_s} c_k^* \ln\left(\frac{c_k^*}{\bar{c}_k}\right) = \sum_{k=1}^{N_s} c_k^* \ln\left(\frac{c_k^*}{c_k^{(N_C)}\mu^{(N_C)}}\right)}{\sum_{k=1}^{N_s} c_k^* \ln(1/c_k^*)} \tag{5}$$

The reason why we divide the KL divergence by the entropy of the target distribution is to enable comparison of the fidelity of the mechanism across target distributions of different complexity. Note that high fidelity corresponds to low values of $F(\boldsymbol{c}^*\|\bar{\boldsymbol{c}})$, vice versa.

Thus, the problem of designing a sequential synthesis mechanism that approximates $\boldsymbol{c}^*$ for a given enzyme specificity $\boldsymbol{\sigma}$, transport rate $\boldsymbol{\mu}$, number of enzymes $N_E$, and number of cisternae $N_C$ is given by

$$Optimization\ A: \quad \bar{D}(\sigma, N_E, N_C, \boldsymbol{c}^*) := \min_{\boldsymbol{\mu},\, \mathbf{M},\, \mathbf{V},\, \mathbf{L}\,\geq\,\mathbf{0}} F(\boldsymbol{c}^*\|\bar{\boldsymbol{c}}), \tag{6}$$

where we emphasize that the optimum fidelity $\bar{D}(\sigma, N_E, N_C, \boldsymbol{c}^*)$ is a function of $(\sigma, N_E, N_C, \boldsymbol{c}^*)$. Note that there is a separation of time scales implicit in Optimization A – the chemical kinetics of the production of glycans and their display on the PM happens over cellular time scales, while the issues of trade-offs and changes of parameters are related to evolutionary timescales.

Optimization A, though well-defined, is a hard problem since the steady-state concentrations (16) are not *explicitly* known in terms of the parameters ($\boldsymbol{\mu}, \mathbf{M}, \mathbf{V}, \mathbf{L}$). In Appendix 4, we formulate an alternative problem *Optimization B* in which the steady-state concentrations are defined explicitly in terms of new parameters $\boldsymbol{\mu}, \mathbf{R}$, and $\mathbf{L}$, and prove that *Optimization A and Optimization B are exactly equivalent*. This is a crucial insight that allows us to obtain all the results that follow. In Appendix 5, we describe the variant of the sequential quadratic programming (SQP) (*Boyd and Vandenberghe, 2004*), which we use to numerically solve the optimization problem.

## Results

The dimension of the optimization search space is extremely large $\approx O(N_s \times N_E \times N_C)$. To make the optimization search more manageable, we make the following simplifying assumptions:

i.  We ignore the $k$-dependence of the vectors ($\mathbf{M}, \mathbf{V}$), or alternatively of $\mathbf{R}$ – see Appendix 4 for details.

ii.  The enzyme-substrate-binding probability $P^{(j)}(k,\alpha)$ is still dependent on the substrate $k$. We assume that the shape function is a scalar (a length), that is, $l_\alpha^{(j)} = \ell_\alpha^{(j)}$. It further simplifies the algebra to assume that the lengths of the substrates are integer multiples of a basic unit (which we take to be 1), that is, $\ell_k = k$. The norm that appears in (3) is taken to be the absolute value

difference $\left| l_k - l_\alpha^{(j)} \right|$. Other metrics, such as $\left| l_k - l_\alpha^{(j)} \right|^2$, corresponding to the elastic distortion model (*Savir and Tlusty, 2007*), do not pose any computational difficulties, and we see that the results of our optimization remain qualitatively unchanged.

iii.  We drop the dependence of the specificity on $\alpha$ and $j$, and take it to be a scalar $\sigma$.

These restrictions significantly reduce the dimension of the optimization search, so much so that in certain limits we can solve the problem analytically (in Appendix 6, we show that *Equation 21* can be solved analytically in the limit $N_s \gg 1$ since the glycan index $k$ can be approximated by a continuous variable, and the recursion relations for the steady-state glycan concentrations *Equations 15–16* can be cast as a matrix differential equation. This allows us to obtain an *explicit* expression for the steady-state concentration in terms of the parameters (**R**, **L**)). This helps us obtain some useful heuristics (Appendix 6) on how to tune the parameters, for example, $N_E$, $N_C$, $\sigma$, and others, in order to generate glycan distributions **c** of a given complexity. These heuristics inform our more detailed optimization using 'realistic' target distributions.

The calculations in Appendix 6 imply, as one might expect, that the synthesis model needs to be more elaborate, that is, needs a larger number of cisternae $N_C$ or a larger number of enzymes $N_E$, in order to produce a more complex glycan distribution. For a real cell type in a niche, the specific elaboration of the synthesis machinery would depend on a variety of control costs associated with increasing $N_E$ and $N_C$. While an increase in the number of enzymes would involve genetic and transcriptional costs, the costs involved in increasing the number of cisternae could be rather subtle.

Notwithstanding the relative control costs of increasing $N_E$ and $N_C$, it is clear from the special case that increasing the number of cisternae achieves the goal of obtaining an accurate representation of the target distribution. Suppose the target distribution $c_k^* = \delta(k - M)$ for a fixed $M \gg 1$, that is, $c_k^* = 1$ when $k = M$, and 0 otherwise, and that the $N_E$ enzymes that catalyze the reactions are highly specific. In this limit, *Optimization A* reduces to a simple enumeration exercise (*Jaiman and Thattai, 2018*): clearly, one needs $N_E = M$ enzymes, one for each $k = 1, \ldots, M$ reactions, in order to generate $\mathcal{P}c_M$. For a single Golgi cisterna with a finite cisternal residence time (finite $\mu$), the chemical synthesis network will generate a significant steady-state concentration of lower index glycans $\mathcal{P}c_k$ with $k < M$, contributing to a low fidelity. To obtain high fidelity, one needs multiple Golgi cisternae with a specific enzyme partitioning $(E_1, E_2, \ldots, E_M)$ with $E_j$ enzymes in cisterna $j = 1, \ldots, N_c$. This argument can be generalized to the case where the target distribution is a finite sum of delta functions. The more general case, where the enzymes are allowed to have variable specificity, needs a more detailed study, to which we turn to next.

## Target distribution from coarse-grained MSMS

As discussed in the section 'Complexity of glycan code', we obtain the target glycan distribution from glycan profiles for real cells using MSMS measurements (*Cummings and Crocker, 2020*). The raw MSMS data, however, is not suitable as a target distribution. This is because it is very noisy, with chemical noise in the sample and Poisson noise associated with detecting discrete events being the most relevant (*Du et al., 2008*). This means that many of the small peaks in the raw data are not part of the signal, and one has to 'smoothen' the distribution to remove the impact of noise.

We use MSMS data from *human* T-cells (*Cummings and Crocker, 2020*) for our analysis. As discussed in the section 'Complexity of glycan code', the GMMs are often used to approximate distributions with a mixed number of modes or peaks (*MacKay, 2003*), or in our setting, a given fixed complexity. Here, we use a variation of the GMMs (see Appendix 1 for details) to create a hierarchy of increasingly complex distributions to approximate the MSMS raw data. Thus, the 3-GMM and 20-GMM approximations represent the low- and high-complexity benchmarks, respectively. In Appendix 1, we show that the likelihood for the glycan distribution of the *human* T-cell saturates at 20 peaks. Thus, statistically the *human* T-cell glycan distribution is accurately approximated by 20 peaks.

This hierarchy allows us to study the trade-off between the complexity of the target distribution and the complexity of the synthesis model needed to generate the distribution as follows. Let $\mathbf{T}^{(i)}$ denote the  -component GMM approximation for the *human* T-cell MSMS data. We sample this target distribution at indices $k = 1, \ldots, N_s$, that represent the glycan indices, and then renormalize to obtain the discrete distribution $\{T_k^{(i)}, k = 1, \ldots, N_s\}$. To highlight the role of target distribution complexity,

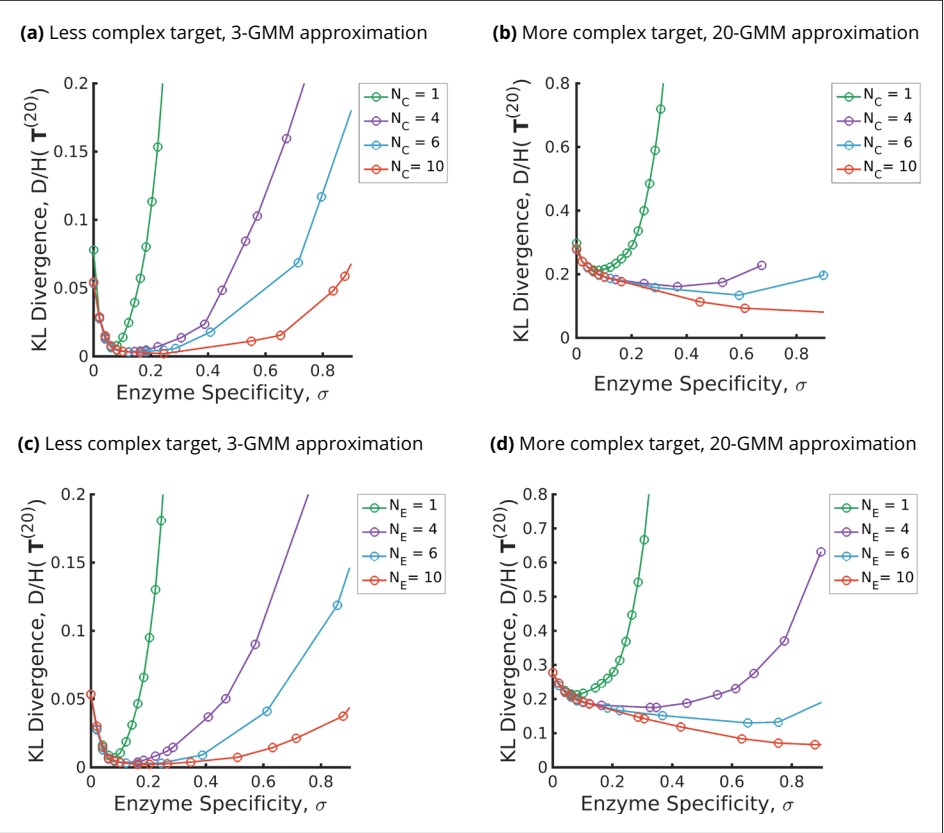

**Figure 3.** Trade-offs amongst the glycan synthesis parameters, enzyme specificity $\sigma$, cisternal number $N_C$, and enzyme number $N_E$ to achieve a complex target distribution $\mathbf{c}^*$. (a, b) Normalised Kullback–Leibler distance $\bar{D}(\sigma, N_E, N_C, \mathbf{c}^*)$ as a function of $\sigma$ and $N_C$ (for fixed $N_E = 3$), (c, d) $\bar{D}(\sigma, N_E, N_C, \mathbf{c}^*)$ as a function of $\sigma$ and $N_E$ (for fixed $N_C = 3$), with the target distribution $\mathbf{c}^*$ set to the 3-Gaussian mixture model (GMM) (less complex) and 20-GMM (more complex) approximations for the *human* T-cell mass spectrometry coupled with determination of molecular structure (MSMS) data. $\bar{D}(\sigma, N_E, N_C, \mathbf{c}^*)$ is a convex function of $\sigma$ for each $(N_E, N_C, \mathbf{c}^*)$, decreasing in $N_C, N_E$ for each $(\sigma, \mathbf{c}^*)$, increasing in the complexity of $\mathbf{c}^*$ for fixed $(\sigma, N_E, N_C)$. The specificity $\sigma_{\min}(\mathbf{c}^*, N_C, N_E) = \mathrm{argmin}_\sigma \{\bar{D}(\sigma, N_E, N_C, \mathbf{c}^*)\}$ that minimizes the error for given $(N_E, N_C, \mathbf{c}^*)$ is an increasing function of $N_C, N_E$ and the complexity of the target distribution $\mathbf{c}^*$.

we focus on the 3-GMM $T^{(3)}$ (low complexity) and 20-GMM approximation $T^{(20)}$ (high complexity) in describing our results.

## Trade-offs between number of enzymes, number of cisternae, and enzyme specificity to achieve given complexity

We summarize the main results that follow from an optimization of the parameters of the glycan synthesis machinery to a given target distribution in *Figure 3* and *Figure 4*.

a. The optimal fidelity $\bar{D}(\sigma, N_E, N_C, \mathbf{c}^*)$ is a convex function of $\sigma$ for fixed values for other parameters (see *Figure 3*), that is, it first decreases with $\sigma$ and then increases beyond a critical value of $\sigma_{\min}$.

The lower complexity distributions can be synthesized with high fidelity with small $(N_E, N_C)$, whereas higher complexity distributions require significantly larger $(N_E, N_C)$ (see *Figure 4a and b*). For a typical mammalian cell, the number of enzymes in the N-glycosylation pathway is in the range $N_E = 10 - 20$ (*Umaña and Bailey, 1997*; *Krambeck and Betenbaugh, 2005*; *Krambeck et al., 2009*; *Fisher and Ungar, 2016*), *Figure 4b* would then suggest that the optimal cisternal number would range from $N_C = 3 - 8$ (*Sengupta and Linstedt, 2011*). The fidelity $\bar{D}(\sigma, N_E, N_C, \mathbf{c}^*)$ is decreasing in $N_C$ and $N_E$ for fixed values of the other parameters, and increasing in the complexity of $\mathbf{c}^*$ for fixed $(\sigma, N_C)$. The marginal contribution

of $N_C$ and $N_E$ in improving fidelity $\bar{D}$ is approximately equal (see **Figure 4a and b**). We discuss the origin of this symmetry later in this section.

b. The optimal enzyme specificity $\sigma_{\min}(\mathbf{c}^*, N_C) = \operatorname{argmin}_\sigma\{\bar{D}(\sigma, \bar{N}_E, N_C, \mathbf{c}^*)\}$, which minimizes the error as function of $(N_C, \mathbf{c}^*)$ with $N_E$ fixed at $\bar{N}_E$, is an increasing function of $N_C$ and the complexity of the target distribution $\mathbf{c}^*$ (**Figure 3a and b** and **Figure 4c and d**). This is consistent with the results in Appendix 6 where we established that the width of the synthesized distribution is inversely dependent on the specificity $\sigma$: since a GMM approximation with fewer peaks has wider peaks, $\sigma_{\min}$ is low, and vice versa. Similar results hold when $N_C$ is fixed at $\bar{N}_C$, and $N_E$ is varied (see **Figure 3c and d** and **Figure 4c and d**).

Our results are consistent with those in **Fisher et al., 2019**. They optimize incoming glycan ratio, transport rate, and effective reaction rates in order to synthesize a narrow target distribution centred around the desired glycan. The ability to produce specific glycans without much heterogeneity is an important goal in the pharmaceutical industry. They define heterogeneity as the total number of glycans synthesized and show that increasing the number of compartments $N_C$ decreases heterogeneity and

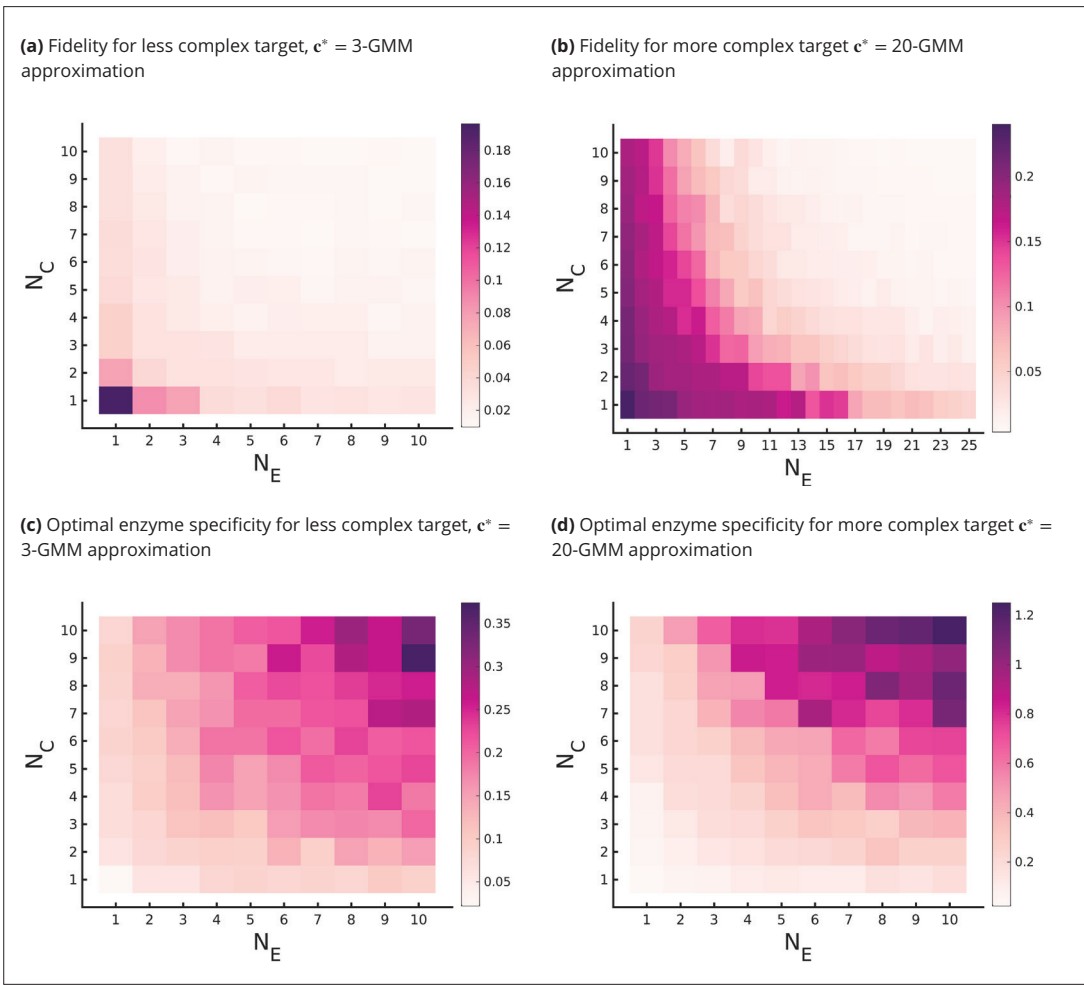

**(a)** Fidelity for less complex target, $\mathbf{c}^* = $ 3-GMM approximation

**(b)** Fidelity for more complex target $\mathbf{c}^* = $ 20-GMM approximation

**(c)** Optimal enzyme specificity for less complex target, $\mathbf{c}^* = $ 3-GMM approximation

**(d)** Optimal enzyme specificity for more complex target $\mathbf{c}^* = $ 20-GMM approximation

**Figure 4.** Fidelity of glycan distribution and optimal enzyme properties to achieve a complex target distribution. The target $\mathbf{c}^*$ is taken from 3-Gaussian mixture model (GMM) (less complex) and 20-GMM (more complex) approximations of the *human* T-cell mass spectrometry coupled with determination of molecular structure (MSMS) data. (**a, b**) Optimum fidelity $\min_\sigma\{\bar{D}(\sigma, N_C, N_E, \mathbf{c}^*)\}$ as a function of $(N_E, N_C)$. More complex distributions require either a larger $N_E$ or $N_C$. The marginal impact of increasing $N_E$ and $N_C$ on the fidelity $\bar{D}$ is approximately equal. (**c, d**) Enzyme specificity $\sigma_{\min}$ that achieves $\min_\sigma\{\bar{D}(\sigma, N_C, N_E, \mathbf{c}^*)\}$ as a function of $(N_E, N_C)$. $\sigma_{\min}$ increases with increasing $N_E$ or $N_C$. To synthesize the more complex 20-GMM approximation with high fidelity requires enzymes with higher specificity $\sigma_{\min}$ compared to those needed to synthesize the broader, less complex 3-GMM approximation.

increases the concentration of the specific glycan. They also show that the effect of compartments in reducing heterogeneity cannot be compensated by changing the transport rate. Our results are entirely consistent with theirs – we have shown that $\bar{D}$ decreases as we increase $N_C$. Thus, if the target distribution has a single sharp peak, increasing $N_C$ will reduce the heterogeneity in the distribution.

We insert an important cautionary note here. It would seem that the results in *Figure 4* imply that there is an approximate $N_E - N_C$ symmetry in the model, that is, increasing either $N_E$ or $N_C$ affects the fidelity, optimal enzyme specificity, and the sensitivity in approximately the same way. This would be an erroneous inference, and is a consequence of the distortion model we have used for calculating the binding probabilities of substrates with enzymes. The root cause for this apparent symmetry is that we have allowed for all enzymes to catalyze reactions in all cisternae (albeit with different efficiencies). This symmetry is violated by simply restricting the activity of the enzymes to be dependent on the cisternae. A simple realization of this in terms of the distortion model is given in Appendix 7.

## Optimal partitioning of enzymes in cisternae

Having studied the optimum $N_E, N_C, \sigma$ to attain a given target distribution with high fidelity, we ask what is the optimal partitioning of the $N_E$ enzymes in these $N_C$ cisternae? Answering this within the context of our chemical reaction model (section 'Chemical reaction and transport network in cisternae') requires some care since it incorporates the following enzymatic features: (a) enzymes with a finite specificity $\sigma$ can catalyze several reactions, although with an efficiency that varies with both the substrate index $k$ and cisternal index $j$, and (b) every enzyme appears in each cisternae; however, their reaction efficiencies depend on the enzyme levels, the enzymatic reaction rates, and the enzyme matching function $\mathbf{L}$, all of which depend on the cisternal index $j$.

Therefore, instead of focusing on the cisternal partitioning of enzymes, we identify the chemical reactions that occur with high propensity in each cisternae. For this we define an effective reaction rate $R_{eff}(j, k)$ for $\mathcal{P}c_k \to \mathcal{P}c_{k+1}$ in the $j$th cisterna as

$$R_{eff}(j, k) = \sum_{\alpha=1}^{N_E} R_\alpha^{(j)} P^{(j)}(k, \alpha). \tag{7}$$

According to our model presented in the section 'Chemical reaction and transport network in cisternae', the list of reactions with high effective reaction rates in each cisterna corresponds to a cisternal partitioning of the perfect enzymes. In a future study, we will consider a Boolean version of a more complex chemical model to address more clearly the optimal enzyme partitioning amongst cisternae.

*Figure 5ai* shows the heat map of the effective reaction rates in each cisterna for the optimal that minimizes the normalized KL distance to the 20-GMM target distribution $\mathbf{T}^{(20)}$ (see *Figure 5aii*). The optimized glycan profile displayed in *Figure 5aiii* is very close to the target. An interesting observation from *Figure 5ai* is that the same reaction can occur in multiple cisternae.

Keeping everything else fixed at the optimal value, we ask whether simply repartitioning the optimal enzymes amongst the cisternae alters the displayed glycan distribution. In *Figure 5bi*, we have exchanged the enzymes of the fourth and second cisterna. The glycan profile after enzyme partitioning (see *Figure 5biii*) is now completely altered (compare *Figure 5bii* with *Figure 5biii*). Thus, one can generate different glycan profiles by repartitioning enzymes amongst the same number of cisternae (*Jaiman and Thattai, 2018*).

## Geometry of the fidelity landscape

Here we show that the optimum solution is not unique, rather it is highly degenerate, with several equally good optimum solutions. Thus the multidimensional fidelity landscape in $\mathbf{R}$, $\boldsymbol{\mu}$, $\mathbf{L}$, and $\sigma$ is typically rugged. We analyse the geometry of this fitness landscape by doing a local Hessian analysis about the optimal solutions.

## Degeneracies in the synthesis model

The synthesis model is highly degenerate, in the sense that many combinations of parameters give rise to the same glycan profile. This makes the optimization non-convex as there are many equally good minima. These degeneracies are both discrete and continuous. The continuous degeneracies correspond to regions in reaction rate ($\mathbf{R}$)-transport rate ($\boldsymbol{\mu}$) space moving along which does not change

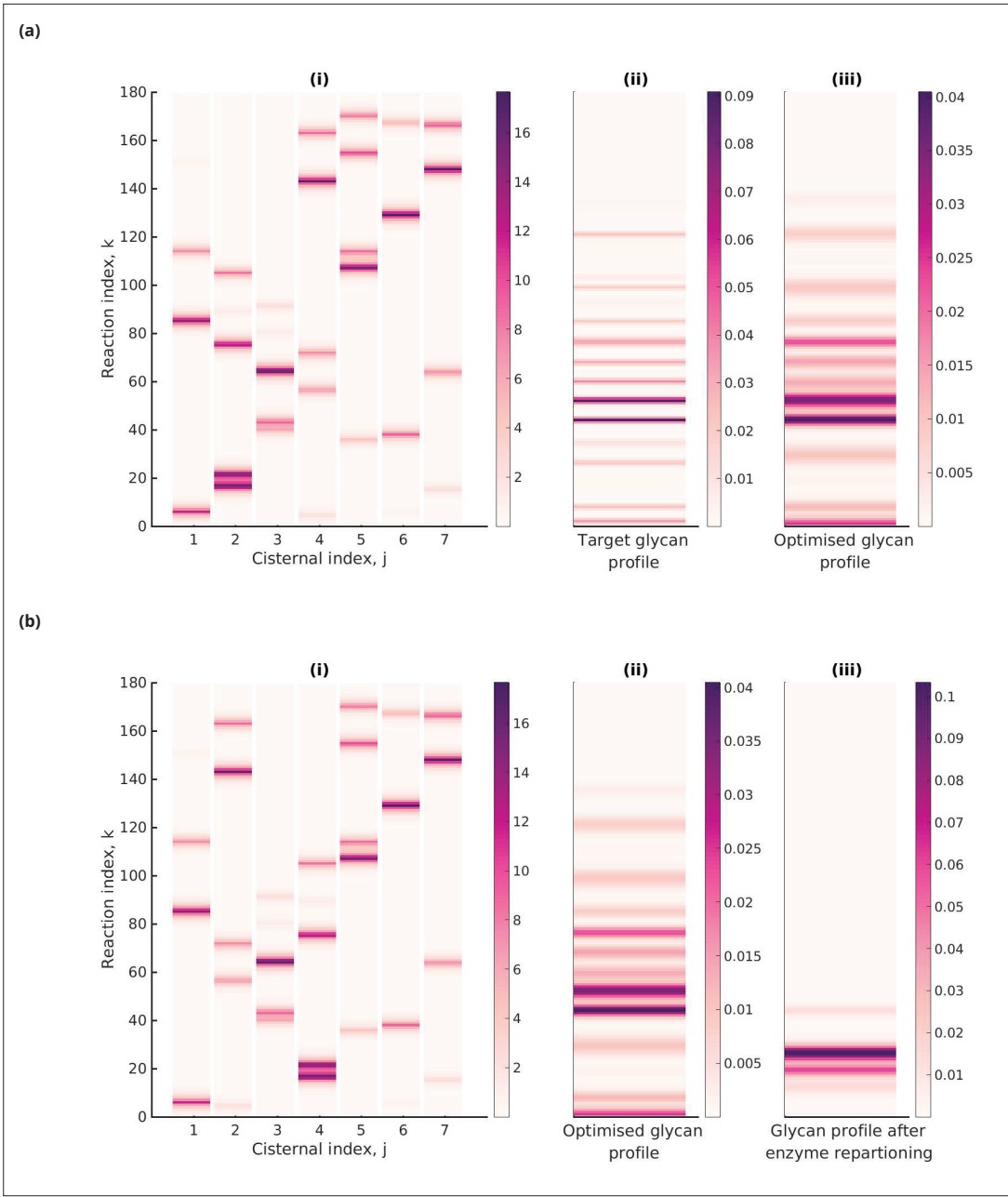

**Figure 5.** Optimal enzyme partitioning in cisternae. (**a**) Heat map of the effective reaction rates in each cisterna (representing the optimal enzyme partitioning) and the steady-state concentration in the last compartment ($\mathbf{c}^{(N_C)}$) for the 20-Gaussian mixture model (GMM) target distribution. Here, $N_E = 5$, $N_C = 7$, normalized $D(\mathbf{T}^{(20)}\|\mathbf{c}^{(N_C)})/H(\mathbf{T}^{(20)}) = 0.11$. (**b**) Effective reaction rates after swapping the optimal enzymes of the fourth and second cisternae. The displayed glycan profile is considerably altered from the original profile.

the concentration profile. The discrete degeneracies are disconnected regions in the parameter space which correspond to the same glycan profile. The number of discrete degeneracies increases exponentially with increase in $(N_E, N_C)$. We also find that the fraction of initial conditions converging to a solution close to the global minima increases on increasing $(N_E, N_C)$. Technical details of these issues are discussed in Appendix 8.

## Stiff and sloppy directions

We analyse the change in fidelity on small perturbations in $\mathbf{R}$, $\boldsymbol{\mu}$, $\mathbf{L}$, and $\sigma$ around the optimal solution. This allows us to determine where the cell needs to develop a tighter control mechanism (*stiff*

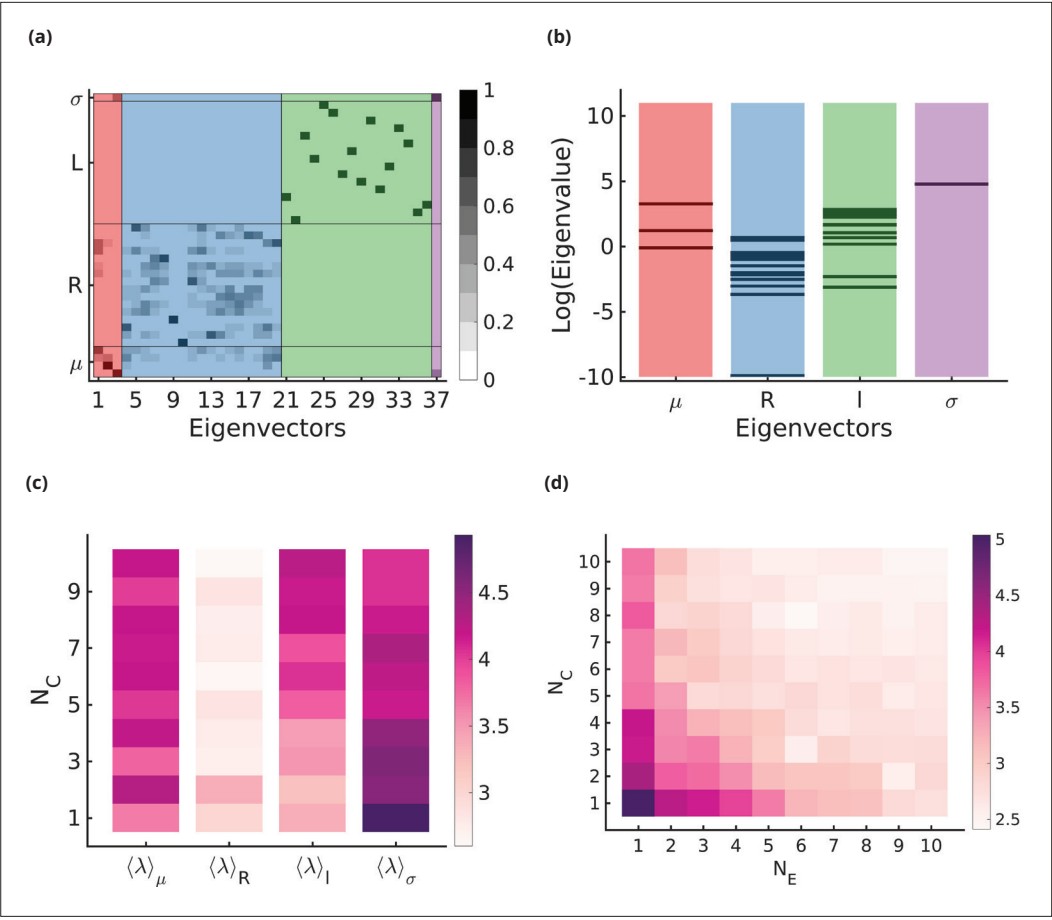

**Figure 6.** Stiff and sloppy directions in the optimization parameters. (**a**) Eigenvectors of the Hessian matrix $\frac{\partial^2}{\partial X_i \partial X_j} F \Big|_{X_{\min}}$ for $(N_E, N_C) = (4, 4)$. The x-axis indexes the $N_C + 2N_E N_C + 1 = 37$ eigenvectors, the y-axis indexes the $N_C + 2N_E N_C + 1$ *components* of the eigenvectors, and the greyscale denotes the absolute value of the component in the range $[0, 1]$. The components are grouped according to $(\mu, R, L, \sigma)$, and the eigenvectors are ordered according to the most dominant component in the eigenvector ($\mu$, orange; $R$, blue; $L$, green; $\sigma$, purple). There is some mixing of the different components ($R$ and $\mu$ or $\sigma$ and $\mu$) but this is usually small. (**b**) The distribution of eigenvalues $\lambda_i$ of the Hessian matrix $\frac{\partial^2}{\partial X_i \partial X_j} F \Big|_{X_{\min}}$. Each stripe represents an eigenvalue, and the location of the stripe on the x-axis represents whether the dominant component of the associated eigenvector belongs to $\mu$, $R$, $L$, or $\sigma$ direction. (**c**) The average stiffness along $\mu$, $R$, $L$, or $\sigma$ directions, defined by the log of the average of eigenvalues corresponding to the eigenvectors in the respective group, as a function of $N_C$ for fixed $N_E = 4$. (**d**) Total average stiffness $\langle \lambda \rangle = \log \left( \frac{\sum \lambda_i}{N_C + 2N_E N_C + 1} \right)$ as a function of $N_E, N_C$.

directions) and where it has more leeway around the optimal values (*sloppy* directions). We do this by analysing the eigenvalues and eigenvectors of the Hessian around the optimal point (details in Appendix 9). We find that small perturbations around the optimal values in $\sigma$ change the glycan profile a lot more compared to perturbations in the other parameters and this stiffness in $\sigma$ generally decreases on increasing $N_E, N_C$ (*Figure 6a–c*). Small perturbations in $\mu$ and some $L$ directions around the optimum also significantly alter the glycan profile and the stiffness increases on increasing $N_C, N_E$, eventually becoming comparable to $\sigma$. The glycan profile is robust to perturbations in most $R$ and some $L$ directions (*Figure 6b*). The total average stiffness of the optimization parameters, defined by the mean of all eigenvalues of the hessian, decreases on increasing $N_E, N_C$ (*Figure 6c*).

## Implications for robustness to parametric noise

Since the synthesized glycan distribution displayed by the cell marks its identity, it must be robust to noise intrinsic to the synthesis machinery. The degeneracy of solutions and sloppy directions in

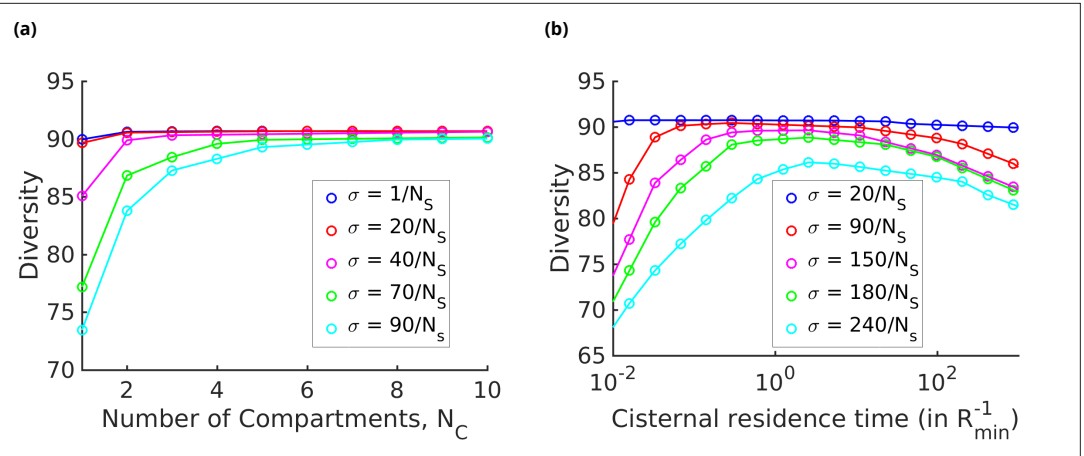

**Figure 7.** Strategies for achieving high glycan diversity. Diversity versus $N_C$ and transport rate $\mu$ at various values of specificity $\sigma$ for fixed $N_E = 3$. (a) Diversity vs. $N_C$ at optimal transport rate $\mu$. Diversity initially increases with $N_C$, but eventually levels off. The levelling off starts at a higher $N_C$ when $\sigma$ is increased. These curves are bounded by the $\sigma = 0$ curve. (b) Diversity vs. cisternal residence time ($\mu^{-1}$) in units of the reaction time ($R_{\min}^{-1}$) at various value of $\sigma$, for fixed $N_C = 4$ and $N_E = 10$.

the fidelity landscape makes the glycan distribution robust to intrinsic noise in the synthesis and cell-to-cell variations in the kinetic parameters. We find that the number of degeneracies increases on increasing ($N_E, N_C$), and the average stiffness of the optimized parameters decreases on increasing ($N_E, N_C$), making the synthesis more robust to parameter fluctuations. Further, while the parameter space is high dimensional, the dimension of *controllable* parameters (measured by the stiff directions) is low dimensional. We find this dimensional reduction a compelling idea which we will take up later.

## Strategies to achieve high glycan diversity

So far we have studied how the complexity of the target glycan distribution places constraints on the evolution of Golgi cisternal number and enzyme specificity. We now take up another issue, namely, how the physical properties of the Golgi cisternae, namely, cisternal number and inter-cisternal transport rate, may drive the diversity of glycans (*Varki, 2011*; *Dennis et al., 2009*). There is substantial correlative evidence to support the idea that cell types that carry out extensive glycan processing employ larger numbers of Golgi cisternae. For example, the salivary Brunner's gland cells secrete mucous that contains heavily O-glycosylated mucin as its major component (*Van Halbeek et al., 1983*). The Golgi complex in these specialized cells contain 9–11 cisternae per stack. Additionally, several organisms such as plants and algae secrete a rather diverse repertoire of large, complex glycosylated proteins, for a variety of functions (*McFarlane et al., 2014*; *Koch et al., 2015*; *O'Neill et al., 2004*; *Hayashi and Kaida, 2011*; *Kumar et al., 2011*; *Gow and Hube, 2012*; *Atmodjo et al., 2013*; *Free, 2013*; *Pauly et al., 2013*; *Burton and Fincher, 2014*). These organisms possess enlarged Golgi complexes with multiple cisternae per stack (*Becker and Melkonian, 1996*; *Mironov et al., 2017*; *Donohoe et al., 2007*; *Mogelsvang et al., 2003*; *Ladinsky et al., 2002*).

We define *diversity* as the total number of glycan species produced above a specified threshold abundance $c_{th}$. This last condition is necessary because very small peaks will not be distinguishable in the presence of noise. In computing the diversity from our chemical synthesis model, we have chosen the threshold to be $c_{th} = 1/N_s$, where $N_s$ is the total number of glycan species. We have checked that the qualitative results do not depend on this choice (see *Appendix 10—figure 1*).

We use the sigmoid function $(1 + e^{-x/\tau})^{-1}$ as a differentiable approximation to the Heaviside function $\Theta(x)$ and define the following optimization to maximize diversity for a given set of parameter values, $N_E, N_C, \sigma$:

$$
\begin{aligned}
\text{Diversity}(\sigma, N_C, N_E) := \quad &\max_{\mu,R,L} \quad \sum_{i=1}^{N_s} \left(1 + e^{-N_s(c_i - c_{th})}\right)^{-1} \\
\text{s.t.} \quad &R_{\min} \leq R_\alpha^{(j)} \leq R_{\max}, \\
&\mu_{\min} \leq \mu^{(j)} \leq \mu_{\max},
\end{aligned}
$$

where, as before, $(\mu_{\max}, \mu_{\min}) = (1, 0.01)/\min$, and $(R_{\max}, R_{\min}) = (20, 0.018)/\min$, and $c_{th} = 1/N_s$ is the threshold. See Appendix 2 for details on the parameter estimation.

The results displayed in **Figure 7a** show that for a fixed specificity $\sigma$ the diversity at first increases with the number of cisternae $N_C$, and then saturates at a value that depends on $\sigma$. For very-high-specificity enzymes, one can achieve very high diversity by appropriately increasing $N_C$. This establishes the link between glycan diversity and cisternal number. However, this link is correlational at best since there are many ways to achieve high glycan diversity – notably by increasing the number of enzymes.

On the other hand, one of the goals of glycoengineering is to produce a particular glycan profile with low heterogeneity (**Fisher et al., 2019**; **Jaiman and Thattai, 2018**). For low-specificity enzymes, the diversity remains unchanged upon increasing the cisternal residence time. For enzymes with high specificity, the diversity typically shows a non-monotonic variation with the cisternal residence time. At small cisternal residence time, the diversity decreases from the peak because of the early exit of incomplete oligomers. At large cisternal residence time, the diversity again decreases as more reactions are taken to completion. Note that the peak is generally very flat, which is consistent with the results in **Fisher et al., 2019**. To get a sharper peak, as advocated for instance by **Jaiman and Thattai, 2018**, one might need to increase the number of high-specificity enzymes $N_E$ further.

## Discussion

The precision of the stereochemistry and enzymatic kinetics of these N-glycosylation reactions (**Varki, 2009**) has inspired a number of mathematical models (**Umaña and Bailey, 1997**; **Krambeck et al., 2009**; **Krambeck and Betenbaugh, 2005**) that predict the N-glycan distribution based on the activities and levels of processing enzymes distributed in the Golgi cisternae of mammalian cells and compare these predictions with N-glycan mass spectrum data. Models such as the KB2005 model (**Umaña and Bailey, 1997**; **Krambeck and Betenbaugh, 2005**; **Krambeck et al., 2009**) are extremely elaborate (with a network of 22,871 chemical reactions and 7565 oligosaccharide structures) and require many chemical input parameters. These models have an important practical role to play, that of being able to predict the impact of the various *chemical parameters* on the glycan distribution, and to evaluate appropriate metabolic strategies to recover the original glycoprofile. Additionally, a recent study by Ungar and coworkers (**Fisher et al., 2019**; **Fisher and Ungar, 2016**) shows how *physical parameters*, such as overall Golgi transit time and cisternal number, can be tuned to engineer a homogeneous glycan distribution. Overall, such models can help predict glycosylation patterns and direct glycoengineering projects to optimize glycoform distributions.

Our focus is different. We are interested in the role of glycans as a marker or molecular code of cell identity (**Gabius, 2018**; **Varki, 2017**; **Pothukuchi et al., 2019**), and in particular, understanding enzymatic and transport processes located in the secretory apparatus of the cell that ensure that this code is generated with high *fidelity*. To do this, we have had to develop a new formal apparatus that allows us to address these questions and discuss trade-offs between competing drives. Since our analysis draws on many diverse fields, we provide a short summary of the assumptions, methods, and results of the article before discussing the implications of our work.

i. The glycan profile on the cell surface is a marker of *cell-type identity* (**Varki, 2009**; **Gabius, 2018**; **Varki, 2017**; **Pothukuchi et al., 2019**). We define the complexity of a glycan profile to be the minimum number of GMM components required to approximate the profile to within the noise floor. We show that with this definition of complexity more complex organisms correlate with higher complexity glycan profiles. We use this to analyse the complexity of the glycan profiles of planaria, hydra, and mammalian cells (**Drickamer and Taylor, 1998**).

ii. The glycans at the cell surface are the end product of a sequential chemical processing via a set of enzymes resident in the Golgi cisternae and transport across cisternae (**Varki, 2017**; **Varki, 1998**; **Pothukuchi et al., 2019**). We have proposed a general model for chemical synthesis and transport that, in principle, allows us to compute the *synthesized* glycan distribution at the cell surface as a function of the enzymes $N_E$, reaction rates $R$, enzyme configurations $L$, specificity of enzymes $\sigma$, number of cisternae $N_C$, and transport rates $\mu$. However the large dimension of the search space makes this optimization intractable. We thus use a simplified synthesis model with fewer parameters; while our quantitative results are based on this simplified model, we believe that at a qualitative level our results have more general validity.

iii. We define the *fidelity* of a synthesis mechanism as the minimum normalized KL divergence (*Cover and Thomas, 2012*; *MacKay, 2003*) between synthesized glycan distribution on the cell surface and a 'target' profile.

iv. The results of the optimization over rates $R$ and enzyme configurations $L$ for a given value of $(N_C, N_E, \sigma)$ and a target distribution $\boldsymbol{c}^*$ of given complexity are given in *Figure 3* and *Figure 4*. Here, we highlight some qualitative consequences of the model:

a. Keeping the number of enzymes fixed, a more elaborate transport mechanism (via control of $N_C$ and $\mu$) is essential for synthesizing high-complexity target distributions to within a high fidelity, or equivalently, low error (*Figure 4a and b*). Fewer cisternae cannot be compensated for by optimizing the enzymatic synthesis via control of parameters $R$, $L$, and $\sigma$. An empirical verification of this would involve a coordinated analysis of the glycan profiles, ultrastructure of Golgi, and the number of glycosylation enzymes across many species.

b. Thus, our study suggests that the requirement that a glycan code of a given complexity be synthesized with sufficiently high fidelity imposes functional control on the Golgi cisternal number. It also provides an argument for the evolutionary requirement of multiple compartments by demonstrating that the fidelity and robustness of the glycan code arising from a chemical synthesis that involves multiple cisternae are higher than the one that involves a single cisterna (keeping everything else fixed) (see *Figure 4a and b* and *Figure 6*) This feature, that with multiple cisternae and precise enzyme partitioning one may generically achieve a highly accurate representation of the target distribution, has been highlighted in an algorithmic model of glycan synthesis *Jaiman and Thattai, 2018*.

Combining (a) and (b), our study quantitatively shows that constructing a high-fidelity representation of a *complex target distribution*, such as those observed in real cells, requires a *complex Golgi machinery* with multiple cisternae, precise enzyme partitioning, and control on enzyme specificity. This definition of fidelity of the glycan code allows us to provide a quantitative argument for the evolutionary requirement of multiple compartments. While it is possible to produce complex glycan distributions in one compartment using a large number of enzymes, such a design would inevitably require a more elaborate genetic cost.

c. Organisms such as plants and algae have a diverse repertoire of glycans that are utilized in a variety of functions (*McFarlane et al., 2014*; *Koch et al., 2015*; *O'Neill et al., 2004*; *Hayashi and Kaida, 2011*; *Kumar et al., 2011*; *Gow and Hube, 2012*; *Atmodjo et al., 2013*; *Free, 2013*; *Pauly et al., 2013*; *Burton and Fincher, 2014*). Our study shows that it is optimal to use low-specificity enzymes to synthesize target distributions with high diversity (*Figure 7*). However, this compromises on the complexity of the glycan distribution, revealing a tension between complexity and diversity. One way of relieving this tension is to have larger $N_E$ and $N_C$.

d. Our study shows that for a fixed $N_C$ and $N_E$, there is an optimal enzyme specificity that achieves the lowest distance from a given target distribution. As we see in *Figure 4d*, this optimal enzyme specificity can be very high for highly complex target distributions. Such high specificity can lower fitness when the environment, and hence the target glycan distribution, fluctuates rapidly, and the synthesis parameters cannot change rapidly enough to track the environment (*Nam et al., 2012*; *Peracchi, 2018*). This compromise, between robustness to a changing environment and high fidelity in synthesizing high-complexity glycan profiles, is achievable by sloppy enzymes coupled with error-correcting mechanisms (*Nam et al., 2012*; *Peracchi, 2018*). However, sloppy enzymes create 'wrong' glycans, and therefore, ex-post error-correcting mechanisms must be in place to correct synthesis errors to ensure high fidelity of the glycan code. A task for the future is to understand the role of intracellular transport in providing non-equilibrium proofreading mechanisms to reduce such coding errors, and its optimal adaptive strategies and plasticity in a time-varying environment.

Combining (c) and (d), we find that keeping the number of enzymes fixed, having low specificity or sloppy enzymes, and larger cisternal number could give rise to a diverse repertoire of functional glycans. Sloppy or promiscuous enzymes bring in the potential for *evolvability* (*Kirschner and Gerhart, 2008*), and sloppiness allows the system to be stable to random mutations in proteins or variations in the target distribution.

e. The model solution is degenerate, in the sense that there are many equally good global minimas. These degeneracies are both continuous and discrete. The continuous degeneracies correspond to regions in the reaction rate – transport rate space, moving along which will not change the concentration profile, thus ensuring *robustness* to internal

noise. This suggests that the distribution is robust to slight cell-to-cell variations in these kinetic parameters.

 f. Our model implies that close to a local minima the inter-cisternal transport rate $\mu$ and the specificity of the enzymes $\sigma$ are stiff directions, that is, the cell should exercise tighter control on $\mu$ and $\sigma$ as compared to the other parameters. The reaction rates close to the local minima are sloppy directions, and moving along these directions does not change the glycan profile much.

 v. Taken together, our quantitative analysis of the trade-offs has deep implications for non-equilibrium self-assembly of the Golgi cisternae, and suggests that the non-equilibrium control of cisternal number must involve a coupling of non-equilibrium self-assembly of cisternae with enzymatic chemical reaction kinetics (*Glick and Malhotra, 1998*).

Admittedly the chemical network that we have considered here is much simpler than the chemical network associated with the possible protein modifications in the secretory pathway. For instance, typical N-glycosylation pathways would involve the glycosylation of a variety of GBPs. Further, apart from N-glycosylation, there are other glycoprotein, proteoglycan, and glycolipid synthesis pathways (*Alberts, 2002*; *Varki, 2009*; *Pothukuchi et al., 2019*). Our task has been to get at a qualitative understanding using quantitative methods and thereby to arrive at general principles. We believe our analysis is generalizable and that the qualitative results we have arrived at would still hold. To conclude, our work establishes the link between the cisternal machinery (chemical and transport) and high-fidelity synthesis of a complex glycan code. We find that the pressure to achieve the target glycan code for a given cell type places strong constraints on the cisternal number and enzyme specificity (*Sengupta and Linstedt, 2011*). An important implication is that a description of the non-equilibrium self-assembly of a fixed number of Golgi cisternae must combine the dynamics of chemical processing and membrane dynamics involving fission, fusion, and transport (*Sengupta and Linstedt, 2011*; *Sachdeva et al., 2016*; *Sens and Rao, 2013*). We believe that this is a promising direction for future research.

## Acknowledgements

We thank M Thattai, A Jaiman, S Ramaswamy, and A Varki for discussions, and S Krishna and R Bhat for very useful suggestions on the manuscript. We thank our group members at the Simons Centre for many incisive inputs. We are very grateful to P Babu and PS Sabarinath for consultations on the MSMS data and literature. MR acknowledges support from the Department of Atomic Energy (India), under project no. RTI4006, the Simons Foundation (grant no. 287975), and a JC Bose Fellowship from DST-SERB (India). We acknowledge the computational facilities at NCBS. This work has received support under the program Investissements d'Avenir launched by the French Government and implemented by ANR with the references ANR-10-LABX-0038 and ANR-10-IDEX-0001-02 PSL. MR thanks Institut Curie for hosting a visit under the Labex program, and QV thanks the Simons Centre (NCBS) for hosting his visit.

## Additional information

### Competing interests

Pierre Sens: Reviewing editor, *eLife*. The other authors declare that no competing interests exist.

### Funding

| Funder | Grant reference number | Author |
|---|---|---|
| Department of Atomic Energy, Government of India | RTI4006 | Madan Rao |
| Simons Foundation | 287975 | Madan Rao |
| JC Bose Fellowship | DST-SERB | Madan Rao |

| Funder | Grant reference number | Author |
|---|---|---|

The funders had no role in study design, data collection and interpretation, or the decision to submit the work for publication.

## Author contributions

Alkesh Yadav, Conceptualization, Data curation, Formal analysis, Investigation, Methodology, Software, Validation, Visualization, Writing - original draft, Writing - review and editing; Quentin Vagne, Investigation, Methodology, Writing - review and editing; Pierre Sens, Funding acquisition, Project administration, Writing - review and editing; Garud Iyengar, Conceptualization, Data curation, Formal analysis, Investigation, Methodology, Project administration, Resources, Software, Supervision, Validation, Visualization, Writing - original draft, Writing - review and editing; Madan Rao, Conceptualization, Data curation, Formal analysis, Funding acquisition, Investigation, Methodology, Project administration, Resources, Software, Supervision, Validation, Visualization, Writing - original draft, Writing - review and editing

## Author ORCIDs

Alkesh Yadav http://orcid.org/0000-0002-4268-8873
Pierre Sens http://orcid.org/0000-0003-4523-3791
Garud Iyengar http://orcid.org/0000-0001-6546-4154
Madan Rao http://orcid.org/0000-0001-6210-6386

## Decision letter and Author response

Decision letter https://doi.org/10.7554/eLife.76757.sa1
Author response https://doi.org/10.7554/eLife.76757.sa2

---

# Additional files

## Supplementary files

• Transparent reporting form

## Data availability

The current manuscript is a computational study, so no data have been generated for this manuscript. The following repository on github contains the code and the data (numerical data + Mass Spec data) that are used in the paper: https://github.com/alkeshyadav/Glycosylation, (copy archived at URL swh:1:rev:a46c6eb76c5f07458d07e44267f48bbaaff6fc5a).

The following dataset was generated:

| Author(s) | Year | Dataset title | Dataset URL | Database and Identifier |
|---|---|---|---|---|
| Yadav A | 2022 | Glycan processing in the Golgi: optimal information coding and constraints on cisternal number and enzyme specificity | https://github.com/alkeshyadav/Glycosylation | GitHub, Glycosylation |

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

## Appendix 1

### Constructing target distributions for glycans of a given cell type

The distribution of the glycans on the cell surface is obtained via mass spectrometry. The x-axis of mass spectroscopy (MS) graphs is mass/charge of the ionized sample molecules, and the y-axis is relative intensity corresponding to each mass/charge value, taking the highest intensity as 100%. This relative intensity roughly correlates with the relative abundances of the molecules in the sample.

The raw MS data is noisy and cannot be directly used as the target distribution in our optimization problem. There are three major sources of noise in the MS data (*Du et al., 2008*): chemical noise in the sample, the Poisson noise associated with detecting discrete events, and the Nyquist–Johnson noise associated with any charge system. We propose a simple model that accounts for the chemical noise and the Poisson sampling noise. Using this noise model and the available MS data, we generate parametric bootstrap samples of glycan measurements and fit a GMM on this sample to approximate the glycan distribution. This GMM probability distribution is used as the target distribution in our numerical experiments.

The MS data obtained from *Cummings and Crocker, 2020*; *Subramanian et al., 2018*; *Sahadevan et al., 2014* had mass ranging between 500 and 5000 Da with intensity reported at every 0.0153 Da. We first bin this MS data into 180 bins and take the maximum value within each bin as the value of intensity for that bin. *Appendix 1—figure 1* plots the raw MS data and the binned distribution. Next, we describe the parametric bootstrap model that we used to generate the glycan data. Let $\bar{I}_k$ represent the relative intensity of the $k$th bin in the binned MS graph. We generate a sample population of glycans using the MS data in the following way:

i. Poisson sampling noise: The MS data does not have absolute count information. We assume an arbitrary maximum count $I_{max}$ and define the intensity $I_k = I_{max}\bar{I}_k$. The plots in *Appendix 1—figure 2a* show that the results are not sensitive to the specific value of $I_{max}$.

ii. Chemical noise: The sample used for MS analysis also contains small amounts of molecules that are not glycans. These appear as very small peaks in the MS data. We assume that the probability $p_k$ that the peak at index $k$ corresponds to a glycan is given by

$$p_k = 1 - e^{-\frac{I_k}{I_{max}}} = 1 - e^{-\bar{I}_k}$$

which adequately suppresses this chemical noise.

iii. Bootstrapped glycan data: The count $n_k$ at the glycan index $k$ is distributed according to the following distribution:

$$\mathrm{n}_k = \begin{cases} 0 & (1 - p_k) & n = 0 \\ n & p_k e^{-I_k}\frac{(I_k)^n}{n!} & n \geq 1. \end{cases}$$

We assume that the MS data was generated from $N$ different cells. Thus, the total count at glycan index $k$ is given by the sum of $N$ i.i.d. samples distributed according to the distribution above. In *Appendix 1—figure 2b*, we show that results are insensitive to $N$. We normalize the count distribution by the total number of counts across all the bins to obtain the bootstrapped probability mass function $PT$.

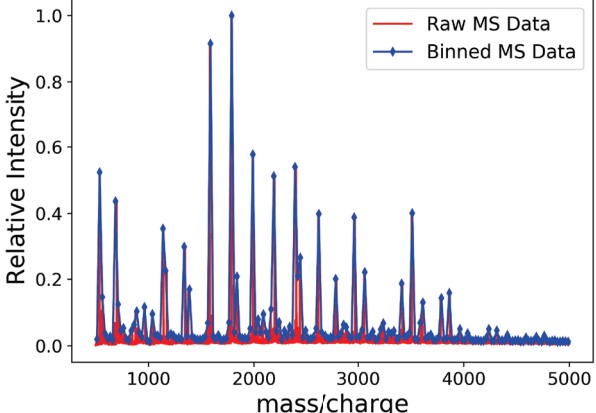

**Appendix 1—figure 1.** The binned mass spectroscopy (MS) data (blue) approximates the raw MS data (red) very well. We use this binned data for Gaussian mixture model (GMM) approximation of the MS data.

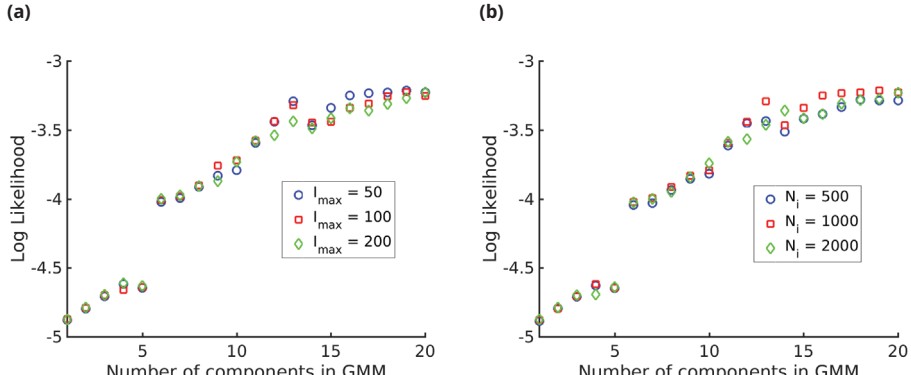

**Appendix 1—figure 2.** Log likelihood vs. number of components ($N$) in the Gaussian mixture model (GMM). We see that the log likelihood saturates at around, $N = 20$, thus 20-GMM is a very good representation of the mass spectroscopy (MS) data from *human* T-cells. The different symbols are for (**a**) different values of the maximum intensity $I_{max} = 50, 100, 200$ and (**b**) different values of the number of i.i.d. samples, $N_i = 500, 1000, 2000$ showing the insensitivity of the log likelihood to the value of $I_{max}$ and. $N_i$

The bootstrapped distribution $p_T$ is noisy, and hence cannot be used directly as the target distribution. We use a GMM-based approach to denoise the raw data. The advantage of using a GMM-based approach is that it creates an easily interpretable hierarchy of increasingly more detailed distributions to approximate the mass spectrometry profile. We define the *complexity* of a mass spectrometry profile as the minimum number of components (individual Gaussians) in the GMM model required to approximate it. The details of the GMM calculations are as follows. We fix the number of components $m$. We want to approximate the bootstrapped probability $p_T$ by the $m$-component mixture of Gaussian distributions $p_{GMM}(\theta) = \sum_{i=1}^{m} w_i \mathcal{N}_{\eta_i, \Delta_i}$, where $\mathcal{N}_{\eta_i, \Delta_i}$ denotes the Gaussian distribution with mean $\eta_i$ and variance $\Delta_i$, $w_i \geq 0$ and $\sum_{i=1}^{m} w_i = 1$, the parameter vector $\theta = (\boldsymbol{w}, \boldsymbol{\eta}, \boldsymbol{\Delta})$ . We compute the optimal $m$-component GMM approximation by minimizing the KL divergence $D(p_T \| p_{GMM}(\boldsymbol{\theta}))$ as a function of parameter vector $\boldsymbol{\theta}$. Since

$$D(p_T \| p_{GMM}(\boldsymbol{\theta})) \quad := \sum_{k=1}^{N_s} p_T(k) \log \left( \frac{p_T(k)}{\sum_i^m w_i \mathcal{N}_{\eta_i, \Delta_i}(k)} \right)$$

$$= \sum_{k=1}^{N_s} p_T(k) \log p_T(k) - \sum_{k=1}^{N_s} p_T(k) \log \left( \sum_i^m w_i \mathcal{N}_{\eta_i, \Delta_i}(k) \right),$$

the optimization problem $\min_\theta D(p_T \| p_{GMM}(\theta))$ is equivalent to

$$\max_\theta g(\theta) := \sum_{k=1}^{N_S} p_T(k) \log \left( \sum_{i=1}^{m} w_i \mathcal{N}_{\eta_i, \Delta_i}(k) \right)$$

This is a non-convex optimization. We use an expectation-maximization (EM)-based iterative heuristic to compute a local maximum. Let $\theta^{(t)}$ denote the current value of the parameters. For each component $i = 1, \ldots, m$, and index $k = 1, \ldots, N_s$, define

$$z_i^{(t)}(k) = \frac{w_i^{(t)} \mathcal{N}_{\eta_i^{(t)}, \Delta_i^{(t)}}(k)}{\sum_{j=1}^{m} w_j^{(t)} \mathcal{N}_{\eta_j^{(t)}, \Delta_j^{(t)}}(k)}.$$

Then $z_i^{(t)}(k) \geq 0$, and $\sum_{i=1}^{m} z_i^{(t)}(k) = 1$. We interpret $z_i^{(t)}(k)$ as the probability that the count in bin $k$ came from component . Define

$$Q(\theta, \theta^{(t)}) = \sum_{k=1}^{N_S} \sum_{i=1}^{m} p_T(k) z_i^{(t)}(k) \log \left( \frac{w_i \mathcal{N}_{\eta_i, \Delta_i}(k)}{z_i^{(t)}(k)} \right)$$

Then, we have that

$$Q(\hat{\theta}, \hat{\theta}) = \sum_{k=1}^{N_S} \sum_{i=1}^{m} p_T(k) \hat{z}_i(k) \log \left( \sum_{i=1}^{m} w_i \mathcal{N}_{\eta_i, \Delta_i}(k) \right) = \sum_{k=1}^{N_S} p_T(k) \log \left( \sum_{i=1}^{m} w_i \mathcal{N}_{\eta_i, \Delta_i}(k) \right) = g(\hat{\theta}),$$

and

$$g(\theta) = \sum_{k=1}^{N_S} p_T(k) \log \left( \sum_{i=1}^{m} \frac{w_i \mathcal{N}_{\eta_i, \Delta_i}(k)}{z_i^{(t)}(k)} z_i^{(t)}(k) \right)$$
$$\geq \sum_{k=1}^{N_S} \sum_{i=1}^{m} p_T(k) z_i^{(t)}(k) \log \left( \frac{w_i \mathcal{N}_{\eta_i, \Delta_i}(k)}{z_i^{(t)}(k)} \right) = Q(\theta, \theta^{(t)}).$$

Define

$$\theta^{(t+1)} = \underset{\theta}{\arg\max} \, Q(\theta, \hat{\theta}) \tag{8}$$

Then, we have that

$$g(\theta^{(t+1)}) \geq Q(\theta^{(t+1)}, \theta^{(t)}) \geq Q(\theta^{(t)}, \theta^{(t)}) = g(\theta^{(t)}).$$

Therefore, the iterative algorithm in (**Equation 11**) generates a sequence $\{\theta^{(t)} : t \geq 1\}$ with non-decreasing values of $g$, and the sequence converges to a local maximum. Next, we show that the optimization in (8) can be computed efficiently.

i.  $w$-update

$$w^{(t+1)} = \arg\max_{w} \sum_{k=1}^{N_S} \sum_{i=1}^{m} p_T(k) z_i^{(t)}(k) \log(w_i) \implies w_i^{(t+1)} = \frac{\sum_{k=1}^{N_S} z_i^{(t)}(k) p_T(k)}{\sum_{i=1}^{m} \sum_{k=1}^{N_S} z_i^{(t)}(k) p_T(k)} \tag{9}$$

ii.  $\eta$-update

$$\eta_i^{(t+1)} = \arg\min_{\eta} \sum_{k=1}^{N_S} p_T(k) \hat{z}_i(k) |k - \eta_i|^2 \implies \eta_i^{(t+1)} = \frac{\sum_{k=1}^{N_S} z_i^{(t)}(k) k}{\sum_{k=1}^{N_S} z_i^{(t)}(k)}. \tag{10}$$

iii.  $\Delta$-update

$$\Delta_i^{(t+1)} = \text{argmax}_{\Delta \geq \Delta_{cut}} \left\{ -\frac{\sum_{k=1}^{N_s} p_T(k) z_i^{(t)}(k) |k - \eta_i^{t+1}|^2}{2\Delta} - \log(\Delta) \right\}$$

$$= \max \left( \sqrt{\frac{\sum_{k=1}^{N_S} p_T(k) z_i^{(t)}(k) |k - \eta_i^{(t+1)}|^2}{\sum_{k=1}^{N_s} p_T(k) z_i^{(t)}(k)}}, \Delta_{cut} \right),$$

(11)

where $\Delta_{cut}$ is the minimum allowed width of the Gaussians, in our case $\Delta_{cut} = 1$ since glycan index, $k \in \{1, 2, \ldots N_S\}$, takes integer values with spacing 1.

Since this is a heuristic algorithm for a non-convex optimization, we performed several initializations of the algorithm to identify the best local maximum.

The number of components $m$ in a GMM is a free parameter. The KL divergence between the true and GMM approximated ($D(p_T \| p_{GMM})$), shown in **Figure 1**, saturates at some value of the number of components, and adding components beyond this only increases model complexity without increasing the quality of approximation. We define the complexity of a mass spectrometry data by the number of components at which saturation is reached. We compare the complexity of glycan profiles of hydra, planaria, and humans. The numbers of cell types in hydra, planaria, and humans are around 41 (**Siebert et al., 2019**), 44 (**Fincher et al., 2018**), and 103 (**Han et al., 2020**), respectively, based on transcriptome analysis (these are lower bounds based on the main cell types, and especially for planaria and hydra, are subject to constant revision). Our analysis of the MSMS data of these organisms suggests that organisms with fewer cell types have less complex glycan distribution.

## Appendix 2

### Parameter estimation

The typical transport time of glycoproteins across the Golgi complex is estimated to be in the range 15–20 mins (*Umaña and Bailey, 1997*), which corresponds to the transport rate $\mu = 0.18/$min. We bound the transport rate for our optimization between 0.01/min and 1/min.

Next, we estimate the range of values for the chemical reaction rates. The injection rate $q$ is in the range 100–1500 pmol/$10^6$ cell 24 hr (*Umaña and Bailey, 1997*; *Krambeck et al., 2009*). For our calculation, we set $q = 387.30$ pmol/$10^6$ cells 24 hr = 0.27 pmol/$10^6$ cells min, where 387.30 is the geometric mean of 100 and 1500. We set the range for the enzymatic rate $R$ to be

$$R_{\min} = \min_{\alpha}\left\{\frac{V_{\max}^{(\alpha)}/\nu}{K_M^{(\alpha)} + \frac{1}{\nu}\frac{q}{\mu}}\right\} \leq R \leq R_{\max} = \max_{\alpha}\left\{\frac{V_{\max}^{(\alpha)}/\nu}{K_M^{(\alpha)}}\right\}.$$

where $K_M^{(\alpha)}$ and $V_{\max}^{(\alpha)}$ denote the Michaelis constants and $V_{\max}$ of the αth enzyme. The conversion from 1 pmol/$10^6$ cells to concentration can be obtained by taking cisternal volume ($\nu$) to be 2.5 μm³ (*Umaña and Bailey, 1997*; *Krambeck et al., 2009*). This gives

$$1 \text{ pmoles}/10^6 \text{ cells} = \frac{10^{-12}\text{moles}}{10^6 \times 2.5 \times 10^{-18} \times 10^3\text{litre}} = 400\mu M. \tag{12}$$

In *Appendix 2—table 1*, we report the parameters for the eight enzymes taken from Table 3 in *Umaña and Bailey, 1997*. From these parameters, it follows that

$$R_{\min} = \min_{\alpha}\left\{\frac{V_{\max}^{(\alpha)}/\nu}{K_M^{(\alpha)} + \frac{1}{\nu}\frac{q}{\mu}}\right\}$$

$$= \frac{V_{\max}^{(7)}/\nu}{K_M^{(7)} + \frac{1}{\nu}\frac{q}{\mu}} = \frac{.16 \times 400\mu M/\text{min}}{3400\mu M + 149.4\mu M} = 0.018\text{min}^{-1}$$

$$R_{\max} = \max_{\alpha}\left\{\frac{V_{\max}^{(\alpha)}/\nu}{K_M^{(\alpha)}}\right\}$$

$$= \frac{V_{\max}^{(1)}/\nu}{K_M^{(1)}} = \frac{5 \times 400\mu M/\text{min}}{100\mu M} = 20\text{min}^{-1}$$

**Appendix 2—table 1.** Enzyme parameters taken from Table 3 in *Umaña and Bailey, 1997* that we use to calculate the bounds on the reaction rate $R$.
Here, $K_M^{(\alpha)}$ and $V_{\max}^{(\alpha)}$ denote the Michaelis constant and $V_{\max}$ of the αth enzyme.

| $\alpha$ | $K_M^{(\alpha)}$(μmol) | $V_{\max}^{(\alpha)}$(pmol/$10^6$ cell-min) |
|---|---|---|
| 1 | 100 | 5 |
| 2 | 260 | 7.5 |
| 3 | 200 | 5 |
| 4 | 100 | 5 |
| 5 | 190 | 2.33 |
| 6 | 130 | .16 |

*Appendix 2—table 1 Continued on next page*

*Appendix 2—table 1 Continued*

| $\alpha$ | $K_M^{(\alpha)}$(μmol) | $V_{max}^{(\alpha)}$(pmol/$10^6$ cell-min) |
|---|---|---|
| 7 | 3400 | .16 |
| 8 | 4000 | 9.66 |

## Appendix 3

### Kinetics of sequential chemical reactions and transport

On including the rates of injection ($q$) and inter-cisternal transport $\mu^{(j)}$ from the cisterna $j$ to $j+1$ into the chemical reaction kinetics, the change in substrate concentrations $c_k^{(j)}$ with time is given by

$$\frac{dc_1^{(1)}}{dt} = q - \sum_{\alpha=1}^{N_E} \frac{V(1,1,\alpha)P^{(1)}(1,\alpha)c_1^{(1)}}{M(1,1,\alpha)\left(1+\sum_{k'=1}^{N_s}\frac{P^{(1)}(k',\alpha)c_{k'}^{(1)}}{M(1,k',\alpha)}\right)} - \mu^{(1)}c_1^{(1)}$$

$$\frac{dc_k^{(1)}}{dt} = \sum_{\alpha=1}^{N_E} \frac{V(1,k-1,\alpha)P^{(1)}(k-1,\alpha)c_{k-1}^{(1)}}{M(1,k-1,\alpha)\left(1+\sum_{k'=1}^{N_s}\frac{P^{(1)}(k',\alpha)c_{k'}^{(1)}}{M(1,k',\alpha)}\right)}$$

$$-\sum_{\alpha=1}^{N_E} \frac{V(1,k,\alpha)P^{(1)}(k,\alpha)c_k^{(1)}}{M(1,k,\alpha)\left(1+\sum_{k'=1}^{N_s}\frac{P^{(1)}(k',\alpha)c_{k'}^{(1)}}{M(1,k',\alpha)}\right)} - \mu^{(1)}c_k^{(1)}$$

(13)

$$\frac{dc_{N_s}^{(1)}}{dt} = \sum_{\alpha=1}^{N_E} \frac{V(1,N_s-1,\alpha)P^{(1)}(N_s-1,\alpha)c_{N_s-1}^{(1)}}{M(1,N_s-1,\alpha)\left(1+\sum_{k'=1}^{N_s}\frac{P^{(1)}(k',\alpha)c_{k'}^{(1)}}{M(1,k',\alpha)}\right)} - \mu^{(1)}c_{N_s}^{(1)}$$

for cisterna-1, and

$$\frac{dc_1^{(j)}}{dt} = \mu^{(j-1)}c_1^{(j-1)} - \sum_{\alpha=1}^{N_E} \frac{V(j,1,\alpha)P^{(j)}(1,\alpha)c_1^{(j)}}{M(j,1,\alpha)\left(1+\sum_{k'=1}^{N_s}\frac{P^{(j)}(k',\alpha)c_{k'}^{(j)}}{M(j,k',\alpha)}\right)} - \mu^{(j)}c_1^{(j)}$$

$$\frac{dc_k^{(j)}}{dt} = \mu^{(j-1)}c_k^{(j-1)} + \sum_{\alpha=1}^{N_E} \frac{V(j,k-1,\alpha)P^{(j)}(k-1,\alpha)c_{k-1}^{(j)}}{M(j,k-1,\alpha)\left(1+\sum_{k'=1}^{N_s}\frac{P^{(j)}(k',\alpha)c_{k'}^{(j)}}{M(j,k',\alpha)}\right)}$$

(14)

$$-\sum_{\alpha=1}^{N_E} \frac{V(j,k,\alpha)P^{(j)}(k,\alpha)c_k^{(j)}}{M(j,k,\alpha)\left(1+\sum_{k'=1}^{N_s}\frac{P^{(j)}(k',\alpha)c_{k'}^{(j)}}{M(j,k',\alpha)}\right)} - \mu^{(j)}c_k^{(j)}$$

$$\frac{dc_{N_s}^{(j)}}{dt} = \mu^{(j-1)}c_{N_s}^{(j-1)} + \sum_{\alpha=1}^{N_E} \frac{V(j,N_s-1,\alpha)P^{(j)}(N_s-1,\alpha)c_{N_s-1}^{(j)}}{M(j,N_s-1,\alpha)\left(1+\sum_{k'=1}^{N_s}\frac{P^{(j)}(k',\alpha)c_{k'}^{(j)}}{M(j,k',\alpha)}\right)} - \mu^{(j)}c_{N_s}^{(j)}$$

for cisternae $j = 2, 3, \ldots, N_C$. These set of dynamical **equations (13)-(14)**, with initial conditions, can be solved to obtain the concentration $c_k^{(j)}(t)$ for $t \geq 0$. **Equations (13)-(14)** automatically obey the conservation law for the protein concentration ($p$), that is, the total protein concentration $p^{(j)} = \sum_{k'=1}^{N_s} c_{k'}^{(j)}$ in the $j$th cisterna automatically satisfies

$$\frac{dp^{(1)}}{dt} = q - \mu^{(1)}p^{(1)}$$

$$\frac{dp^{(j)}}{dt} = \mu^{(j-1)}p^{(j-1)} - \mu^{(j)}p^{(j)}$$

for $j = 2, 3, \ldots N_C$.

At steady state, the left-hand side of **equations (13)-(14)** is set to zero, which, after rescaling the kinetic parameters in terms of the injection rate $q$, that is, $V(j,k,\alpha) = V(j,k,\alpha)/q$ and $\mu^{(j)} = \mu^{(j)}/q$, gives

the following recursion relations for the steady-state concentrations of the glycans in each cisterna. In the first cisterna,

$$c_1^{(1)} = \frac{1}{\mu^{(1)} + \sum_{\alpha=1}^{N_E} \frac{V(1,1,\alpha)P^{(1)}(1,\alpha)c_1^{(1)}}{M(1,1,\alpha)\left(1+\sum_{k'=1}^{N_s} \frac{P^{(1)}(k',\alpha)c_{k'}^{(1)}}{M(1,k',\alpha)}\right)}}$$

$$c_k^{(1)} = \frac{\sum_{\alpha=1}^{N_E} \frac{V(1,k-1,\alpha)P^{(1)}(k-1,\alpha)c_{k-1}^{(1)}}{M(1,k-1,\alpha)\left(1+\sum_{k'=1}^{N_s} \frac{P^{(1)}(k',\alpha)c_{k'}^{(1)}}{M(1,k',\alpha)}\right)}}{\mu^{(1)} + \sum_{\alpha=1}^{N_E} \frac{V(1,k,\alpha)P^{(1)}(k,\alpha)c_k^{(1)}}{M(1,k,\alpha)\left(1+\sum_{k'=1}^{N_s} \frac{P^{(1)}(k',\alpha)c_{k'}^{(1)}}{M(1,k',\alpha)}\right)}} \tag{15}$$

$$c_{N_s}^{(1)} = \frac{\sum_{\alpha=1}^{N_E} \frac{V(1,N_s-1,\alpha)P^{(1)}(N_s-1,\alpha)c_{N_s-1}^{(1)}}{M(1,N_s-1,\alpha)\left(1+\sum_{k'=1}^{N_s} \frac{P^{(1)}(k',\alpha)c_{k'}^{(1)}}{M(1,k',\alpha)}\right)}}{\mu^{(1)}}$$

and in cisternae $j \geq 2$,

$$c_1^{(j)} = \frac{\mu^{(j-1)}c_1^{(j-1)}}{\mu^{(j)} + \sum_{\alpha=1}^{N_E} \frac{V(j,1,\alpha)P^{(j)}(1,\alpha)c_1^{(j)}}{M(j,1,\alpha)\left(1+\sum_{k'=1}^{N_s} \frac{P^{(j)}(k',\alpha)c_{k'}^{(j)}}{M(j,k',\alpha)}\right)}}$$

$$c_k^{(j)} = \frac{\mu^{(j-1)}c_k^{(j-1)} + \sum_{\alpha=1}^{N_E} \frac{V(j,k-1,\alpha)P^{(j)}(k-1,\alpha)c_{k-1}^{(j)}}{M(j,k-1,\alpha)\left(1+\sum_{k'=1}^{N_s} \frac{P^{(j)}(k',\alpha)c_{k'}^{(j)}}{M(j,k',\alpha)}\right)}}{\mu^{(j)} + \sum_{\alpha=1}^{N_E} \frac{V(j,k,\alpha)P^{(j)}(k,\alpha)c_k^{(j)}}{M(j,k,\alpha)\left(1+\sum_{k'=1}^{N_s} \frac{P^{(j)}(k',\alpha)c_{k'}^{(j)}}{M(j,k',\alpha)}\right)}} \tag{16}$$

$$c_{N_s}^{(j)} = \frac{\mu^{(j-1)}c_{N_s}^{(j-1)} + \sum_{\alpha=1}^{N_E} \frac{V(j,N_s-1,\alpha)P^{(j)}(N_s-1,\alpha)c_{N_s-1}^{(j)}}{M(j,N_s-1,\alpha)\left(1+\sum_{k'=1}^{N_s} \frac{P^{(j)}(k',\alpha)c_{k'}^{(j)}}{M(j,k',\alpha)}\right)}}{\mu^{(j)}}$$

*Equations (15)-(16)* automatically imply that the total steady-state glycan concentration in each cisterna $j = 1, \ldots, N_c$ is given by

$$\sum_{k=1}^{N_s} c_k^{(j)} = \frac{1}{\mu^{(j)}}.$$

## Appendix 4

### Reformulation of Optimization A

Define a new set of parameters,

$$R(j, k, \alpha) = \sum_{\alpha=1}^{N_E} \frac{V(j, k, \alpha)}{M(j, k, \alpha) \left(1 + \sum_{k'=1}^{N_s} \frac{P^{(j)}(k', \alpha) c_{k'}^{(j)}}{M(j, k', \alpha)}\right)} \tag{17}$$

where $\mathbf{c}$ denotes the steady-state glycan concentration corresponding to a specific $(\boldsymbol{\mu}, \mathbf{M}, \mathbf{V}, \mathbf{L})$. Define $\mathbf{v}$ by the following set of linear equations:

$$v_1^{(1)} = \frac{1}{\mu^{(1)} + \sum_{\alpha=1}^{N_E} R(1, 1, \alpha) P^{(1)}(1, \alpha)}$$

$$v_k^{(1)} = \frac{v_{k-1}^{(1)} \sum_{\alpha=1}^{N_E} R(1, k-1, \alpha) P^{(1)}(k-1, \alpha)}{\mu^{(1)} + \sum_{\alpha=1}^{N_E} R(1, k, \alpha) P^{(1)}(k, \alpha)} \tag{18}$$

$$v_{N_s}^{(1)} = \frac{v_{N_s-1}^{(1)} \sum_{\alpha=1}^{N_E} R(1, N_s - 1, \alpha) P^{(1)}(N_s - 1, \alpha)}{\mu^{(1)}}$$

for $j = 1$, and

$$v_1^{(j)} = \frac{v_1^{(j-1)} \mu^{(j-1)}}{\mu^{(j)} + \sum_{\alpha=1}^{N_E} R(j, 1, \alpha) P^{(j)}(1, \alpha)}$$

$$v_k^{(j)} = \frac{v_k^{(j-1)} \mu^{(j-1)}}{\mu^{(j)} + \sum_{\alpha=1}^{N_E} R(j, k, \alpha) P^{(j)}(k, \alpha)}$$

$$+ \frac{v_{k-1}^{(j)} \sum_{\alpha=1}^{N_E} R(j, k-1, \alpha) P^{(j)}(k-1, \alpha)}{\mu^{(j)} + \sum_{\alpha=1}^{N_E} R(j, k, \alpha) P^{(j)}(k, \alpha)} \tag{19}$$

$$v_{N_s}^{(j)} = \frac{v_{N_s}^{(j-1)} \sum_{\alpha=1}^{N_E} R(j, N_s - 1, \alpha) P^{(j)}(N_s - 1, \alpha)}{\mu^{(j)}} + \frac{v_{N_s}^{(j-1)} \mu^{(j-1)}}{\mu^{(j)}}$$

for $j = 2, \ldots, N_C$. Then, by the definition of $\mathbf{R}$ in (17), it trivially follows that the steady-state concentration $\mathbf{c}$ corresponding to $(\boldsymbol{\mu}, \mathbf{M}, \mathbf{V}, \mathbf{L})$ is a solution for (18)–(19).

Next, we show that for $\mathbf{v}$ obtained from (18)–(19) for any parameter $(\boldsymbol{\mu}, \mathbf{R}, \mathbf{L})$, there exists parameter $(\boldsymbol{\mu}, \mathbf{M}, \mathbf{V}, \mathbf{L})$ such that (15)–(16) are automatically satisfied when we set $\mathbf{c} = \mathbf{v}$, that is, $\mathbf{v}$ is the steady-state concentration for $(\boldsymbol{\mu}, \mathbf{M}, \mathbf{V}, \mathbf{L})$, and vice versa. Let

$$\mathcal{A} = \left\{ [c_k^{(j)}]_{j,k} : \begin{array}{l} \mu^{(j)} \geq 0, M(j, k, \alpha) \geq 0, V(j, k, \alpha) \geq 0, l_\alpha^{(j)} \geq 0, \\ [c_k^{(j)}]_{jk} \text{ given by (15) and (16),} \end{array} \right\}$$

and let

$$\mathcal{B} = \left\{ [v_k^{(j)}]_{j,k} : \begin{array}{l} \mu^{(j)} \geq 0, R(j, k, \alpha) \geq 0, l_\alpha^{(j)} \geq 0 \\ [v_k^{(j)}]_{j,k} \text{ given by (18) and (19)} \end{array} \right\}.$$

Then, our task is to show that $\mathcal{A} = \mathcal{B}$. Suppose $[c_k^{(j)}]_{j,k} \in \mathcal{A}$. Let $[M(j, k, \alpha)]$, $[V(j, k, \alpha)]$, and $[l_\alpha^{(j)}]$ be the corresponding parameters. Define

$$R(j,k,\alpha) = \sum_{\alpha=1}^{N_E} \frac{V(j,k,\alpha)}{M(j,k,\alpha)\left(1 + \sum_{k'=1}^{N_s} \frac{P^{(j)}(k',\alpha)c_{k'}^{(j)}}{M(j,k',\alpha)}\right)} \geq 0$$

Then $[c_k^{(j)}]_{j,k} \in \mathcal{B}$.

Next, suppose $[v_k^{(j)}]_{j,k} \in \mathcal{B}$. Let $[R(j,k,\alpha)]$, $[l_\alpha^{(j)}]$ denote the corresponding parameters. Since $\sum_{k=1}^{N_s} v_k^{(j)} = 1/\mu^{(j)} < \infty$, it follows that $\sum_{k=1}^{N_s} P^{(j)}(k,\alpha)v_k^{(j)} < 1/\mu^{(j)} < \infty$. Thus, there exist parameters $[M(j,k,\alpha)]$, $[V(j,k,\alpha)]$, and $[l_\alpha^{(j)}]$ such that

$$R(j,k,\alpha) = \sum_{\alpha=1}^{N_E} \frac{V(j,k,\alpha)}{M(j,k,\alpha)\left(1 + \sum_{k'=1}^{N_s} \frac{P^{(j)}(k',\alpha)v_{k'}^{(j)}}{M(j,k',\alpha)}\right)} \tag{20}$$

Therefore, $[v_k^{(j)}]_{j,k}$ satisfy (15) and (16), that is, $[v_k^{(j)}]_{j,k} \in \mathcal{A}$.

Thus, the set of all concentration profiles defined by (18)–(19) as a function of all possible values of the parameters $(\boldsymbol{\mu}, \mathbf{R}, \mathbf{L})$ is identical to the set defined by (15)–(16) as a function of $(\boldsymbol{\mu}, \mathbf{M}, \mathbf{V}, \mathbf{L})$. This is a crucial insight since it allows us to search the entire parameter space using (18)–(19), where the concentration is known explicitly in terms of $(\boldsymbol{\mu}, \mathbf{R}, \mathbf{L})$.

To pose the new optimization problem, it is convenient to define $\bar{v}_i^{(j)} = \mu^{(j)} v_i^{(j)}$, for $j = 1, \ldots, N_C$, and $i = 1, \ldots, N_s$. Furthermore, the glycan distribution of the surface is given by $\mu^{N_C} v_i^{N_C} = \bar{v}_i$. Thus, it follows that *Optimization B*

$$\bar{D}(\sigma, N_E, N_C, \mathbf{c}^*) := \min_{\boldsymbol{\mu}, \mathbf{R}, \mathbf{L} \geq \mathbf{0}} F(\mathbf{c}^* \| \bar{\mathbf{v}}) \tag{21}$$

is equivalent to (6). Since $\mathbf{v}$ is explicitly known as a function of $(\mathbf{R}, \mathbf{L})$, Optimization B (21) is a more tractable optimization problem than (6). Note that in this setting the function $D(\sigma, N_E, N_C, \mathbf{c}^*)$ in (21) is independent of the rates $\boldsymbol{\mu}$. See *Appendix 4—figure 1* for a flow chart of the two optimization schemes.

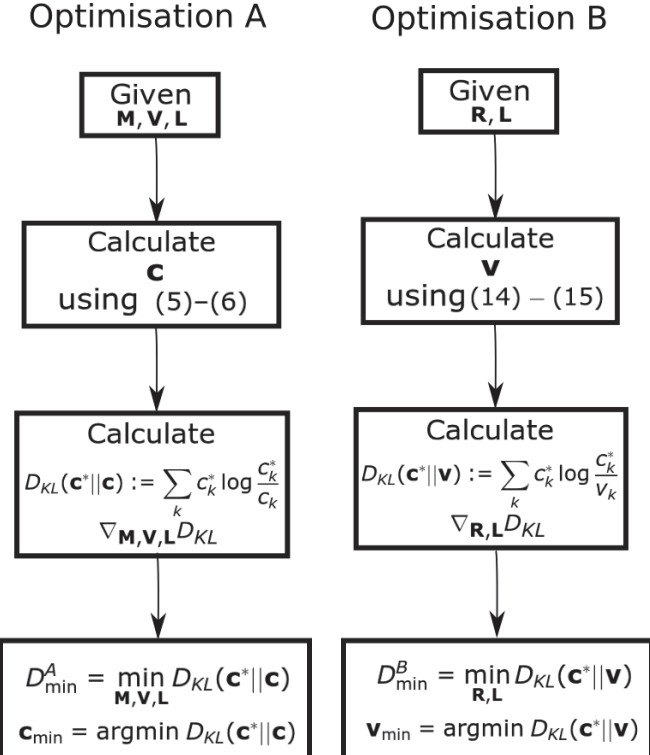

**Appendix 4—figure 1.** Flow chart showing the optimization schemes for Optimization A and B. We prove that $D^A_{\min} = D^B_{\min}$ by showing the set of all **c** is equal to the set of all **v**. We additionally establish that the optimum $\mathbf{v}_{\min} = \mathbf{c}_{\min}$.

While 21 is easy to implement, we note that the parameters (e.g. reaction rates, specificity) are not constrained to take only physically relevant values; a legitimate concern is that the absence of such physicochemical constraints might drive this optimization to physically unrealistic solutions.

There are two possible ways to impose these parameter constraints. One is to impose constraints on the 'microscopic' chemical parameters, such as the rate of individual reactions $R(j, k, \alpha)$ and the inter-cisternal transport rate $\mu^{(j)}$. These take into consideration constraints arising from molecular enzymatic processes. The other is to impose constraints on 'global' physical parameters, such as the total transport time across the Golgi cisternae and the average enzymatic reaction time. Here, we impose constraints on the microscopic reaction and transport parameters to define constrained *Optimization B*:

$$\bar{D}(\sigma, N_C, N_E, \mathbf{c}^*) := \quad \min_{\boldsymbol{\mu}, \mathbf{R}, \mathbf{L}} \quad F(\mathbf{c}^* \| \bar{\mathbf{v}})$$
$$\text{s.t.} \quad R_{\min} \le R(j, k, \alpha) \le R_{\max},$$
$$\mu_{\min} \le \mu^{(j)} \le \mu_{\max}$$
$$1 \le \ell^{(j)}_\alpha \le N_S.$$

The upper and lower bounds on the rates **R** and $\boldsymbol{\mu}$ are estimated in Appendix 2: $\mu_{\max} = 1/\text{min}$ (resp. $\mu_{\min} = .01/\text{min}$) and $R_{\max} = 20/\text{min}$ (resp. $R_{\min} = .018/\text{min}$).

## Appendix 5

### Numerical scheme for performing the non-convex optimization

We solve Optimization C using the numerical scheme detailed below. The optimization problem consists of minimizing a non-convex objective with linear box constraints. We use the MATLAB FMINCON function to solve this optimization. We use SQP, a gradient-based iterative optimization scheme for solving optimizations with non-linear differentiable objective and constraints. Since our problem is non-convex and SQP only gives local minima, we initialize the algorithm with many random initial points. We use SOBOLSET function of MATLAB to generate space filling pseudo-random numbers. We have taken 1000 initializations for each $N_E, N_C$, and $\sigma$ value. We have taken 50 equally spaced points between 0 and 1 to explore the $\sigma$-space for *Figure 3*. Some minor fluctuations in $D$ due to non-convexity of the objective function in the final results were smoothed out by taking the convex hull of the $D$ vs. $\sigma$ graph. The results for $\sigma_{min}(N_E, N_C)$ and $D(\sigma_{min}, N_E, N_C)$ (*Figure 4*) were obtained by adding $\sigma$ to the optimization vector and then performing the optimization.

A similar numerical scheme was used to optimize diversity.

## Appendix 6

### Analytical solution when $N_s \gg 1$

It is possible to obtain analytical expressions for the steady-state glycan distribution in the limit $N_s \gg 1$ when the glycan index $k$ can be approximated by a continuous variable. In this case, (15)–(16) can be cast as differential equations,

$$\frac{dc_k^{(1)}}{dk} \approx c_k^{(1)} - c_{k-1}^{(1)}$$

$$= \left( \frac{\sum_{\alpha=1}^{N_E} R(1, k-1, \alpha) \exp(-\sigma|k-1-l_\alpha^{(1)}|)}{\mu^{(1)} + \sum_{\alpha=1}^{N_E} R(1, k, \alpha) \exp(-\sigma|k-l_\alpha^{(1)}|)} - 1 \right) c_{k-1}^{(1)} \tag{22}$$

$$\approx - \left( \frac{\mu^{(1)} + \frac{d}{dk} \sum_{\alpha=1}^{N_E} R(1, k, \alpha) \exp(-\sigma|k-l_\alpha^{(1)}|)}{\mu^{(1)} + \sum_{\alpha=1}^{N_E} R(1, k, \alpha) \exp(-\sigma|k-l_\alpha^{(1)}|)} \right) c_k^{(1)},$$

and

$$\frac{dc_k^{(j)}}{dk} \approx c_k^{(j)} - c_{k-1}^{(j)}$$

$$= \frac{\mu^{(j-1)}}{\mu^{(j)} + \sum_{\alpha=1}^{N_E} R(j, k, \alpha) \exp(-\sigma|k-l_\alpha^{(j)}|)} c_k^{(j-1)} \tag{23}$$

$$- \left( \frac{\mu^{(j)} + \frac{d}{dk} \sum_{\alpha=1}^{N_E} R(j, k, \alpha) \exp(-\sigma|k-l_\alpha^{(j)}|)}{\mu^{(j)} + \sum_{\alpha=1}^{N_E} R(j, k, \alpha) \exp(-\sigma|k-l_\alpha^{(j)}|)} \right) c_k^{(j)}$$

for $j = 2, \ldots, N_C$. In **equation 22** and **equation 23**,

$$\frac{d}{dk} \sum_{\alpha=1}^{N_E} R(j, k, \alpha) \exp(-\sigma|k-l_\alpha^{(j)}|)$$

$$= \sum_{\alpha=1}^{N_E} R(j, k, \alpha) \sigma \exp(-\sigma|k-l_\alpha^{(j)}|)(1 - 2\mathbb{I}(k \geq l_\alpha)) + R'(j, k, \alpha) \exp(-\sigma|k-l_\alpha^{(j)}|) \tag{24}$$

where the indicator function $\mathbb{I}(\cdot)$ is equal to 1 if the argument is true, and 0 otherwise and $R'(j, k, \alpha)$ is the derivative of $R(j, k, \alpha)$ with respect to $k$.

Define a vector function $C(k) \in \mathbb{R}_c^N$ of the continuous variable $k$ by $C(k) = [c_k^{(1)}, c_k^{(2)}, \ldots c_k^{(N_C)}]$. Then, (**equation 22**) and (**equation 23**) can be written as

$$\frac{dC(k)}{dk} = M(k)C(k) \tag{25}$$

where the matrix $M(k)$ is given by

$$M(k) = \begin{bmatrix} A^{(1)}(k) & 0 & 0 & 0 & \ldots 0 \\ B^{(2)}(k) & A^{(2)}(k) & 0 & 0 & \ldots 0 \\ 0 & B^{(3)}(k) & A^{(3)}(k) & 0 & \ldots 0 \\ \vdots & \vdots & \vdots & \vdots & \vdots \\ 0 & \ldots & 0 & B^{(N_C)}(k) & A^{(N_C)}(k) \end{bmatrix} \tag{26}$$

with

$$A^{(j)}(k) = -\frac{\mu^{(j)} + \frac{d}{dk}\sum_{\alpha=1}^{N_E} R(j,k,\alpha)\exp(-\sigma|k - l_\alpha^{(j)}|)}{\mu^{(j)} + \sum_{\alpha=1}^{N_E} R(j,k,\alpha)\exp(-\sigma|k - l_\alpha^{(j)}|)}$$

$$B^{(j)}(k) = \frac{\mu^{(j-1)}}{\mu^{(j)} + \sum_{\alpha=1}^{N_E} R(j,k,\alpha)\exp(-\sigma|k - l_\alpha^{(j)}|)}$$

The functions $A^{(j)}(k)$ and $B^{(j)}(k)$ involve absolute value and indicator functions; therefore, the differential equation has to be solved in a piecewise manner assuming continuity of solution $C(k)$.

The general solution of (25)

$$C(k) = C_0 \exp\left(\Omega(k)\right) \tag{27}$$

is written in terms of the Magnus function $\Omega(k) = \sum_{n=1}^{\infty} \Omega(n,k)$, obtained from the Baker–Campbell–Hausdorff formula (**Blanes et al., 2009**),

$$\Omega(1,k) = \int_0^k M(k_1)dk_1$$

$$\Omega(2,k) = \frac{1}{2}\int_0^k dk_1 \int_0^{k_1} dk_2 \left[M(k_1), M(k_2)\right]$$

$$\Omega(3,k) = \frac{1}{6}\int_0^k dk_1 \int_0^{k_1} dk_2 \int_0^{k_2} dk_3 \left[M(k_1), \left[M(k_2), M(k_3)\right]\right] + \left[M(k_3), \left[M(k_2), M(k_1)\right]\right]$$

$$\cdots\cdots$$

where $\left[M(k_1), M(k_2)\right] := M(k_1)M(k_2) - M(k_2)M(k_1)$ is the commutator, and the higher order terms in ... contain higher order nested commutators.

Here, we establish conditions under which the series $\sum_{n=1}^{\infty} \Omega(n,k)$ that defines solution $C(k)$ to the differential *equation (25)* converges. We also solve (25) for some special cases.

The commutator

$$[M(k_1), M(k_2)] = \begin{bmatrix} 0 & 0 & 0 & 0 & \ldots & 0 \\ a_{21} & 0 & 0 & 0 & \ldots & 0 \\ a_{31} & a_{32} & 0 & 0 & \ldots & 0 \\ 0 & a_{42} & a_{43} & 0 & \ldots & 0 \\ \vdots & \vdots & \vdots & \vdots & & \vdots \\ 0 & \ldots & & a_{n,n-2} & a_{n,n-1} & 0 \end{bmatrix}$$

where

$$a_{i,i-1} = A^{(i-1)}(k_2)B^{(i)}(k_1) + A^{(i)}(k_1)B^{(i)}(k_2) - A^{(i-1)}(k_1)B^{(i)}(k_2) + A^{(i)}(k_2)B^{(i)}(k_1)$$

$$a_{i,i-2} = B^{(i-1)}(k_2)B^{(i)}(k_1) - B^{(i-1)}(k_1)B^{(i)}(k_2)$$

The general form of $\Omega(n,k)$ is given by **Blanes et al., 2009**

$$\Omega(n,k) = \frac{z_n}{n!}\int_0^k dk_1 \int_0^{k_1} dk_2 \ldots \int_0^{k_{n-2}} dk_{n-1} \int_0^{k_{n-1}} dk_n \sum_l W_l M(k_{p_1^l})M(k_{p_2^l})\ldots M(k_{p_n^l}) \tag{28}$$

where $(p_1^{(l)}, p_2^{(l)} \ldots p_n^{(l)})$ is a permutation of $(1,2,3,\ldots n)$, $W_l \in \{-1,1\}$, and $z_n \in 1,\ldots.n$.

Let $\bar{A} = \max_{k,l,m} |M_{l,m}(k)|$. Define

$$= \begin{bmatrix} \bar{A} & 0 & 0 & 0\ldots & 0 \\ \bar{A} & \bar{A} & 0 & 0\ldots & 0 \\ 0 & \bar{A} & \bar{A} & 0\ldots & 0 \\ \vdots & \vdots & \vdots & & \vdots \\ 0 & \ldots & 0 & \bar{A} & \bar{A} \end{bmatrix}$$

We can bound all the matrix elements of $\Omega(n,k)$ in the following way:

$$\Omega_{lm}(n,k) \quad \leq z_n \bar{M}_{l,m}^n \int_0^k dk_1 \int_0^{k_1} dk_2 \ldots \int_0^{k_{n-1}} dk_n$$

$$= z_n \bar{M}^n \Big|_{lm} \frac{k^n}{n!} \tag{29}$$

The matrix

$$^n = \begin{bmatrix} a_{11} & 0 & 0 & 0 & \ldots & 0 \\ a_{21} & a_{22} & 0 & 0 & \ldots & 0 \\ a_{31} & a_{32} & a_{33} & 0 & \ldots & 0 \\ a_{41} & a_{42} & a_{43} & a_{44} & \ldots & 0 \\ \vdots & \vdots & \vdots & \vdots & & \vdots \\ a_{n1} & \ldots & & a_{n,n-2} & a_{n,n-1} & a_{nn} \end{bmatrix}$$

where $a_{lm} = S_{lm}(n)\bar{A}^n$ for appropriately defined polynomials $S_{l,m}(n)$. Thus, it follows that $\Omega_{lm} \leq z_n S_{lm}(n)(A^*)^n \frac{k^n}{n!}$ and $\Omega_{l,m}(k) \leq \sum_{n=1}^{\infty} z_n S_{l,m}(n)(A^*)^n \frac{k^n}{n!}$. Consequently, the series will converge if $\bar{A}k < 1$, that is, $k \leq \frac{1}{\bar{A}}$. Assuming $\mu^{(j)} = \mu \; \forall j$, we can bound $\bar{A}$ as

$$\bar{A} \leq \max_{j,k} \left( \frac{\mu + \sigma \sum_{\alpha=1}^{N_E} R(j,k,\alpha)\exp(-\sigma|k - l_\alpha^{(j)}|)}{\mu + \sum_{\alpha=1}^{N_E} R(j,k,\alpha)\exp(-\sigma|k - l_\alpha^{(j)}|)} + \frac{\sum_{\alpha=1}^{N_E} R'(j,k,\alpha)\exp(-\sigma|k - l_\alpha^{(j)}|)}{\mu + \sum_{\alpha=1}^{N_E} R(j,k,\alpha)\exp(-\sigma|k - l_\alpha^{(j)}|)} \right) \tag{30}$$

Since the parameters $\mu$, $\sigma$, $R(j,k,\alpha)$, $l_\alpha^{(j)}$, and $N_E$ are finite and positive, and $R'(j,k,\alpha)$ is finite, $\bar{A}$ has a finite upper bound, implying that $k$ is always greater than zero, and the series has a finite radius of convergence.

While in principle we can obtain the glycan profile for any $N_E$ and $N_C$ with arbitrary accuracy, assuming $R(j,k,\alpha) = R_\alpha^{(j)}$, we provide explicit formulae for a few representative cases: (i) $(N_E = 1, N_C = 1)$ and (ii) $(N_E = 1, N_C = 2)$.

i. $N_E = 1, N_C = 1$: The solution of the differential equation is given by

$$c(k) = \begin{cases} c_0 e^{-k} \left( \frac{\mu + R\exp(-\sigma(l-k))}{\mu + R\exp(-\sigma l)} \right)^{(1/\sigma)-1} & k \leq l \\ c(l)e^{-(k-l)} \left( \frac{\mu + R}{\mu + R\exp(-\sigma(k-l))} \right)^{(1/\sigma)+1} & k > l \end{cases} \tag{31}$$

A representative concentration profile is plotted in *Appendix 6—figure 1*. The concentration profile consists of two distinct components: an initial exponential decay, and then an exponential rise and fall concentrated around $l$. The relative weight of these two components is controlled by the sensitivity $\sigma$ and the rate $R$. Such explicit formulae can be obtained for any $N_E > 1$, as long as $N_C = 1$.

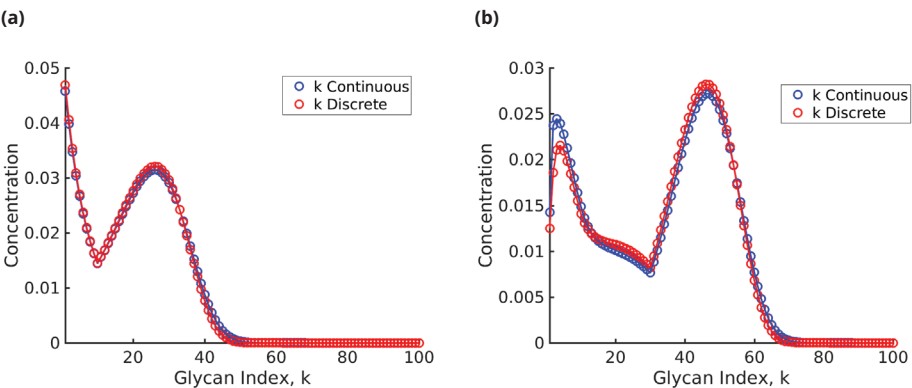

**Appendix 6—figure 1.** Glycan concentration profile calculated from the model using (**a**) formula (31) for $N_E = N_C = 1$ and (**b**) formulae (32)–(36) for $N_E = 1, N_C = 2$.

ii. $N_E = 1, N_C = 2$: The concentration profile $c^{(2)}$ in cisterna-2 can be obtained from the following calculation. Let $l^{(j)}$ denote the 'length' of the enzyme in cisterna $j = 1, 2$. For $k \leq \min\{l^{(1)}, l^{(2)}\}$,

$$c^{(2)}(k) = c_0 \mu^{(1)} e^{-k} \left( \frac{\mu^{(2)} + R^{(2)} \exp(-\sigma(l^{(2)} - k))}{\mu^{(1)} + R^{(1)} e^{-\sigma l^{(1)}}} \right)^{(1/\sigma)-1} \int_0^k \frac{(\mu^{(1)} + R^{(1)} \exp(-\sigma(l^{(1)} - k)))^{(1/\sigma)-1}}{(\mu^{(2)} + R^{(2)} \exp(-\sigma(l^{(2)} - k)))^{1/\sigma}} dk$$

$$+ c^{(2)}(0) e^{-k} \left( \frac{\mu^{(2)} + R^{(2)} e^{-\sigma(l^{(2)} - k)}}{\mu^{(2)} + R^{(2)} e^{-\sigma l^{(2)}}} \right)^{(1/\sigma)-1} \tag{32}$$

Next, consider the case where $l^{(1)} \leq l^{(2)}$. Then, for $l^{(1)} < k \leq l^{(2)}$

$$c^{(2)}(k) = c^{(1)}(l^{(1)}) \mu^{(1)} e^{-(k-l^{(1)})} (\mu^{(1)} + R^{(1)})^{(1/\sigma)+1} (\mu^{(2)} + R^{(2)} \exp(-\sigma(l^{(2)} - k)))^{(1/\sigma)-1}$$

$$\int_{l^{(1)}}^k \frac{(\mu^{(2)} + R^{(2)} \exp(-\sigma(l^{(2)} - k)))^{-1/\sigma}}{(\mu^{(1)} + R^{(1)} \exp(-\sigma(k - l^{(1)})))^{(1/\sigma)+1}} dk \tag{33}$$

$$+ c^{(2)}(l^{(1)}) e^{-(k-l^{(1)})} \left( \frac{\mu^{(2)} + R^{(2)} e^{-\sigma(l^{(2)} - k)}}{\mu^{(2)} + R^{(2)} e^{-\sigma(l^{(2)} - l^{(1)})}} \right)^{(1/\sigma)-1}$$

and for $l^{(1)} \leq l^{(2)} < k$,

$$c^{(2)}(k) = c^{(1)}(l^{(1)}) \mu^{(1)} e^{-(k-l^{(1)})} \left( \frac{\mu^{(1)} + R^{(1)}}{\mu^{(2)} + R^{(2)} \exp(-\sigma(k - l^{(2)}))} \right)^{(1/\sigma)+1}$$

$$\int_{l^{(2)}}^k \frac{(\mu^{(2)} + R^{(2)} \exp(-\sigma(k - l^{(2)})))^{1/\sigma}}{(\mu^{(1)} + R^{(1)} \exp(-\sigma(k - l^{(1)})))^{(1/\sigma)+1}} dk \tag{34}$$

$$+ c^{(2)}(l^{(2)}) e^{-(k-l^{(2)})} \left( \frac{\mu^{(2)} + R^{(2)}}{\mu^{(2)} + R^{(2)} e^{-\sigma(k - l^{(2)})}} \right)^{(1/\sigma)+1}$$

Next, the case where $l^{(1)} \geq l^{(2)}$. For $l^{(2)} < k \leq l^{(1)}$,

$$c^{(2)}(k) \quad = c_0 \mu^{(1)} e^{-k} \frac{(\mu^{(1)}+R^{(1)}e^{-\sigma l^{(1)}})^{1-(1/\sigma)}}{(\mu^{(2)}+R^{(2)}\exp(-\sigma(k-l^{(2)})))^{(1/\sigma)+1}} \int_{l^{(2)}}^{k} \frac{(\mu^{(1)}+R^{(1)}\exp(-\sigma(l^{(1)}-k)))^{(1/\sigma)-1}}{(\mu^{(2)}+R^{(2)}\exp(-\sigma(k-l^{(2)})))^{-1/\sigma}}\,dk$$

$$+c^{(2)}(l^{(2)})e^{l^{(2)}-k}\left(\frac{\mu^{(2)}+R^{(2)}}{\mu^{(2)}+R^{(2)}e^{-\sigma(k-l^{(2)})}}\right)^{(1/\sigma)+1}$$

(35)

For $^{(2)} \le l^{(1)} < k$,

$$c^{(2)}(k) \quad = c^{(1)}(l^{(1)})\mu^{(1)}e^{-(k-l^{(1)})}\left(\frac{\mu^{(1)}+R^{(1)}}{\mu^{(2)}+R^{(2)}\exp(-\sigma(k-l^{(2)}))}\right)^{(1/\sigma)+1}$$

$$\int_{l^{(2)}}^{k} \frac{(\mu^{(2)}+R^{(2)}\exp(-\sigma(k-l^{(2)})))^{1/\sigma}}{(\mu^{(1)}+R^{(1)}\exp(-\sigma(k-l^{(1)})))^{(1/\sigma)+1}}\,dk$$

(36)

$$+c^{(2)}(l^{(1)})e^{-(k-l^{(1)})}\left(\frac{\mu^{(2)}+R^{(2)}e^{-\sigma(l^{(1)}-l^{(2)})}}{\mu^{(2)}+R^{(2)}e^{-\sigma(k-l^{(2)})}}\right)^{(1/\sigma)+1}$$

The integrals in (32)–(36) can be evaluated numerically. The result of the numerical computation is shown in *Appendix 6—figure 1*.

*Appendix 6—figure 2a–d* plots the glycan profile $c_k$ *vs.* $k$ as one varies the enzyme specificity $\sigma$, the reaction rates $R$, and transport rates $\mu$, for two different values of $N_E$ and $N_C$. The results in the plots lead us to the following general observations:

i.  Very-low-specificity enzymes cannot generate complex glycan distributions. Keeping everything else fixed, intermediate or high-specificity enzymes can generate glycan distributions of higher complexity by increasing $N_E$ or $N_C$ (*Appendix 6—figure 2a and c*).
ii. Decreasing the specificity $\sigma$ or increasing the rates $R$ increases the proportion of higher index glycans. Keeping everything else fixed, changes in the rate $R$ have a stronger impact on the relative weights of the higher index glycans to lower index glycans. The relative weight of the higher index glycans increases with increasing $N_E$ and $N_C$ (*Appendix 6—figure 2b–d*).
iii. Keeping everything else fixed, decreasing enzyme specificity increases the spread of the distribution around the peaks (*Appendix 6—figure 2a and c*).

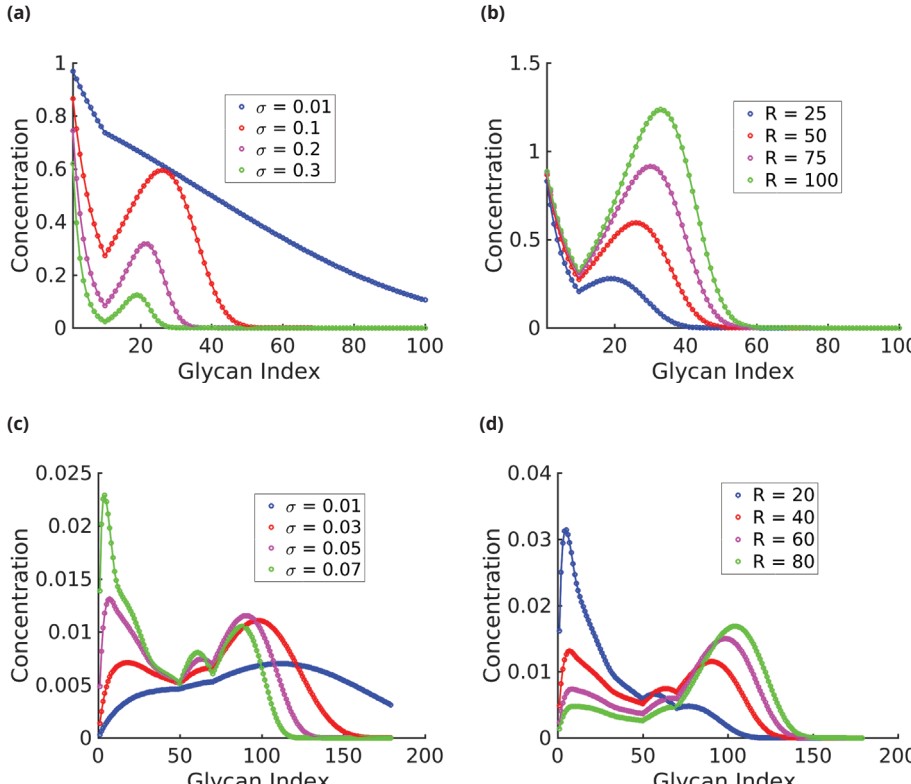

**Appendix 6—figure 2.** Glycan profile $\{c_k : k = 1, \ldots, N_s\}$ as a function of specificity $\sigma$ (**a**, **c**) and reaction rates $R$ (**b**, **d**). (**a**) $N_E = N_C = 1, (R = 50, \mu = 1, l = 10)$. $c_k$ decreases exponentially with $k$ for very low and very high $\sigma$; however, the decay rate is lower at low $\sigma$. For intermediate values of $\sigma$, the distribution has *exactly* two peaks, one of which is at $k = 0$, and eventually decays exponentially. The width of the distribution is a decreasing function of $\sigma$. (**b**) $N_E = N_C = 1, (\sigma = 0.1, \mu = 1, l = 10)$. At low $R$, $c_k$ is concentrated at low $k$. The proportion of higher index glycans in an increasing function of $R$. (**c**) $N_E = N_C = 2, (R = 40, \mu = 1, [l_1^{(1)}, l_2^{(1)}, l_1^{(2)}, l_2^{(2)}] = [10, 30, 50, 70])$. As $\sigma$ increases, the distribution becomes more complex – from a single-peaked distribution at low $\sigma$ to a maximum of four-peaked distribution at high $\sigma$. The peaks gets sharper, and more well defined as $\sigma$ increases. (**d**) $N_E = N_C = 2, (R = 40, \mu = 1, [l_1^{(1)}, l_2^{(1)}, l_1^{(2)}, l_2^{(2)}] = [10, 30, 50, 70])$. As in the plots in (**b**), increasing $R$ shifts the peaks towards higher index glycans and the proportion of higher index glycan increases.

## Appendix 7

### Extended distortion model shows lack of apparent symmetry

The results in *Figure 4* seem to imply that there is an approximate $N_E - N_C$ symmetry in the model, that is, increasing either $N_E$ or $N_C$ affects the fidelity, optimal enzyme specificity, and the sensitivity in approximately the same way. This is a consequence of the distortion model we are using for calculating the binding probabilities of substrates with enzymes, which allows every enzyme $\alpha$ to in principle catalyze any reaction in any cisterna. This allowed for the ideal enzyme length $l_\alpha^{(j)}$ in *Equation 3* to vary across the cisternae in an unconstrained manner, leading to simplification in the calculation. We now find that by changing this aspect of the model the apparent symmetry between $N_E - N_C$ is lifted. A more reasonable model for the ideal enzyme length is given by

$$\ell_\alpha^{(j)} = \ell_\alpha^{(0)} + \delta\ell_\alpha^{(j)}, \qquad \delta\ell_\alpha^{(j)} \in [-\ell_b^{(j)}, \ell_b^{(j)}],$$

that is, the nominal length $\ell_\alpha^{(0)}$ can be distorted in a cisterna by a correction $\delta\ell_\alpha^{(j)}$ but within a specified bound $\ell_b^{(j)}$ that is not subject to optimization. One can render some enzymes inactive in certain cisternae by choosing appropriate values of $\ell_\alpha^{(0)}$ and $\delta\ell_\alpha^{(j)}$. For small values for the bound $\ell_b$, for example, $l_b/N_s \leq 0.2$ (here, $N_s - 1$ is the number of enzymatic reactions), the decrease in $\bar{D}$ on increasing $N_C$ is small compared to increasing $N_E$ (see *Appendix 7—figure 1*). On the other hand, for large $\ell_b$, for example, $l_b/N_S \geq 0.3$, there is an approximate symmetry between $N_E$ and $N_C$ (see *Appendix 7—figure 1*). Here, we have taken the bounds to be compartment independent, that is, $\ell_b^{(j)} = \ell_b$.

**(a)** $l_b/N_S = 0.01$ **(b)** $l_b/N_S = 0.1$ **(c)** $l_b/N_S = 0.5$

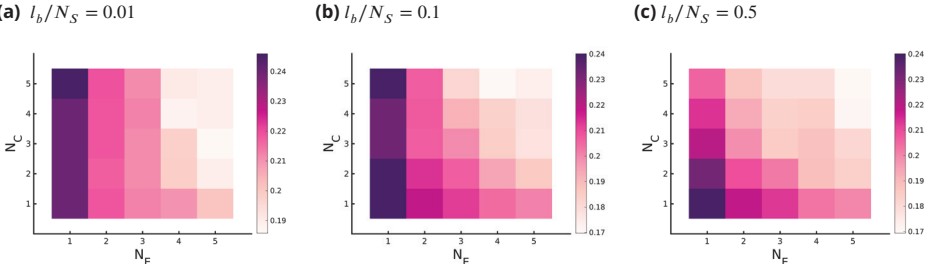

**Appendix 7—figure 1.** Optimum fidelity $\bar{D}_{KL}$ as a function of $(N_E, N_C)$ for different values of $\ell_b/N_s$, where $\ell_b$ bounds the deformation in the ideal length $\ell_\alpha^{(0)}$ of an enzyme $\alpha = 1, \ldots, N_E$. Small values of $\ell_b$ restrict all enzymes from working in all cisternae and all substrates, where large value of $\ell_b$ removes this constraint.

# Appendix 8

## Redundancies and non-convexity of the optimization

### Validation of the numerical optimization scheme

In order to test whether our numerical optimization procedure is able to converge to the global minimum, we run the following test. We generate 100 random values of $(\mu, \boldsymbol{R}, \boldsymbol{L}, \sigma)$ within their respective ranges for a problem instance with $(N_E = 2, N_C = 2)$. The sampled value for $(\mu, \boldsymbol{R}, \boldsymbol{L}, \sigma)$ is used to generate concentration profiles that are then used as the target distribution for the optimization. Since the target distribution is achievable, the optimal value of the constrained *Optimization B* for these sampled targets is $\bar{D} = 0$. We solve the constrained *Optimization B* using our numerical scheme. The average optimal value $\bar{D}$ across all sampled values was 9.1835e-07, 30 out of 100 values were exactly zero, and the highest $\bar{D}$ was 1.1761e-05. Therefore, the optimization scheme was able to recover the concentration profiles almost exactly. Next, we ask whether the optimization problem recovers the value of $(\mu, \boldsymbol{R}, \boldsymbol{L}, \sigma)$ that was used to create the particular target distribution. We were able to recover $\sigma$ exactly, except in cases where the concentration profile was almost a delta function at the first glycan (see *Appendix 8—figure 1*). This is because $\sigma$ decides the typical width of the empirical distribution, and hence the optimal $\sigma$ is determined by the typical width of the target distribution, except in the pathological case of a concentration profile that is almost a delta function at the first glycan – such a concentration profile can be made produced for any value $\sigma$ by simply making transport $\mu$ very fast as compared to the reaction rates.

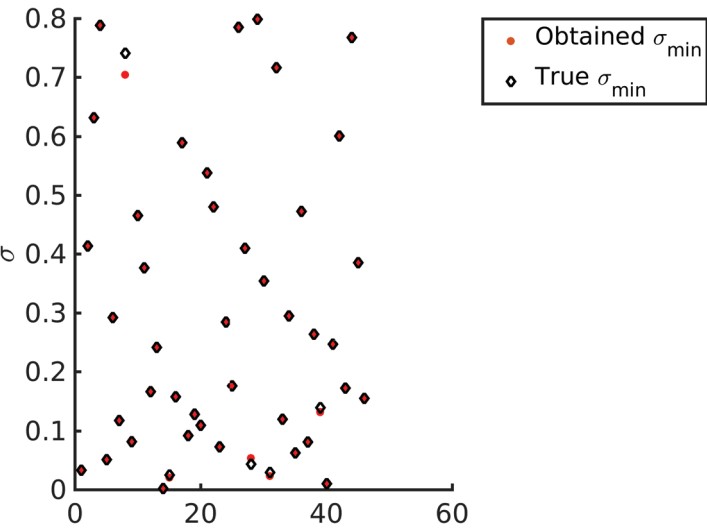

**Appendix 8—figure 1.** Recovering the $\sigma$ values for different target distribution. Note that barring four data points all other optimized $\sigma$ values (red dots) exactly overlap with the corresponding target $\sigma$ (diamonds).

We note that the optimization in $(\mu, \boldsymbol{R}, \boldsymbol{L})$ is not convex and leads to many equally good minimas corresponding to different values of $(\mu, \boldsymbol{R}, \boldsymbol{L})$. The resulting redundancies in the model and their importance are discussed next.

### Degeneracy in the model

Recall that in *equation 7*, we defined

$$R_{eff}(j, k) = \frac{\sum_\alpha R_\alpha^{(j)} \exp(-\sigma |k - l_\alpha^{(j)}|)}{\mu^{(j)}}$$

In terms of these renormalized rates, the steady-state glycan concentration can be written as

$$\bar{c}_k^{(j)} = \frac{R_{eff}(j, k)\bar{c}_{k-1}^{(j)} + \bar{c}_k^{j-1}}{1 + R_{eff}(j, k)}, \tag{37}$$

that is, the concentration is only a function of $R_{eff}(k, j)$. Thus, any combination of $(\boldsymbol{\mu}, \boldsymbol{R}, \boldsymbol{L}, \sigma)$ that maps to the same value of $R_{eff}$ will result in the same concentration profile and will be indistinguishable from the perspective of the objective function. Additionally, the mapping from $\boldsymbol{R}_{eff}$ to the concentration profile $\bar{c}$ also has degeneracy. We show these redundancies in the schematic below, which shows a systematic reduction in dimension to 1 (scalar), which is the quantity we optimize,

$$
\begin{array}{ccccccc}
\mathbf{R}, \mathbf{L}, \boldsymbol{\mu}, \sigma & \longrightarrow & \mathbf{R}_{eff} & \longrightarrow & \bar{c} & \longrightarrow & F \\
N_C + 2N_E N_C + 1 & & N_C(N_S - 1) & & N_S - 1 & & 1
\end{array}
$$

Since $F(\boldsymbol{c}_T \| \bar{\boldsymbol{c}}) = 0$ if and only if $\boldsymbol{c}_T = \bar{\boldsymbol{c}}$, it follows that the last mapping does not have redundancy. Some of the sources of degeneracies in the mapping from $(\boldsymbol{\mu}, \boldsymbol{R}, \boldsymbol{L}, \sigma)$ to $\boldsymbol{R}_{eff}$ are as follows:

i.   For fixed $(\sigma, \boldsymbol{L})$, setting $R_\alpha^{(j)} \leftarrow \gamma R_\alpha^{(j)}$ and $\mu_j \leftarrow \gamma^{-1} \mu_j$ leaves $R_{eff}$ invariant.
ii.  Permutations in the $\alpha$ index leave $R_{eff}$ invariant. Thus, there are at least $(N_E!)^{N_C}$ distinct minima that map to the same value of $R_{eff}$, and therefore, the same concentration $\bar{c}$.

Additionally, there are degeneracies coming from the optimization which depend on the target distribution $\boldsymbol{c}_T$. Having discussed the sources of degeneracies of the optimized solution, we now discuss the distribution of the optimized solutions.

## Distribution of minima

To study the behaviour of the optimization algorithm for different initial points, we numerically investigate the distribution of function values at different local minima. Since the dimension of the optimization problem is $N_C + 2N_E N_C + 1$, which is large, we divide the optimization space into a grid of $I = n_p^{(N_C + 2N_E N_C + 1)}$ points. We did this numerical experiment for $(N_E, N_C) \in \{(1, 1), (1, 2), (2, 1), (2, 2)\}$. The value of $n_p = 3$ for $(N_E, N_C) = (2, 2)$ and $n_p = 4$ for the rest. The target distribution for all the cases is a single Gaussian with mean 20, standard deviation 5, with support on $1 \leq k \leq 20$. The results of this numerical experiment are summarized in *Appendix 8—figure 2* and *Appendix 8—table 1*, from which we deduce the following:

i.   A large fraction of the initial starting points converge to a set of degenerate minima with objective function value exactly equal to the global minimum. These minima are a result of the degeneracies of the optimization problem.
ii.  There are other local minima with objective value very close to (but not equal) the global minimum. Most initial points converge to one of these two sets of minima.
iii. Finally, there are a small set of local minima with significantly higher objective values. These correspond to minima with $\sigma = 0$. The fraction of initial points that converge to such minima reduces as the dimension of the optimization space increases.

**Appendix 8—table 1.** Distribution of local minima.

| $N_E$ | $N_C$ | min $\bar{D}_{KL}$ | max $\bar{D}_{KL}$ | Fraction of initial conditions within $\bar{D}_{KL} \leq 0.0228$ |
|---|---|---|---|---|
| 1 | 1 | 0.0228 | 0.44 | 0.56 |
| 2 | 1 | 0.0081 | 0.44 | 0.73 |
| 1 | 2 | 0.0051 | 0.29 | 0.70 |
| 2 | 2 | 1.17e-4 | 0.29 | 0.84 |

**(a)** $N_E = 1, \; N_C = 1$

**(b)** $N_E = 2, \; N_C = 1$

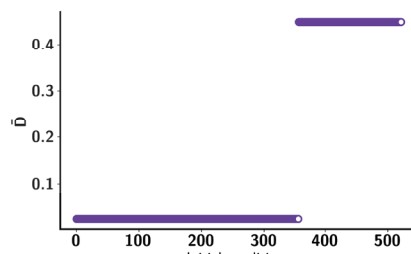

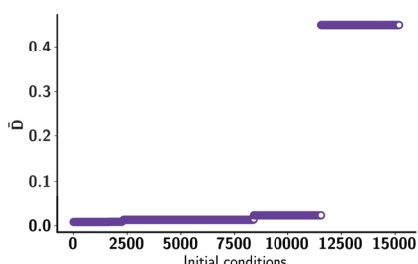

**(c)** $N_E = 1, \; N_C = 2$

**(d)** $N_E = 2, \; N_C = 2$

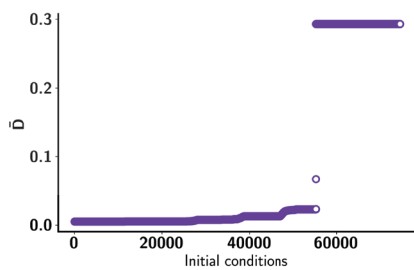

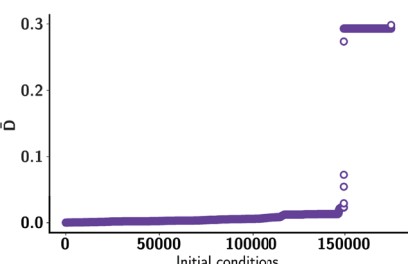

**Appendix 8—figure 2.** $\bar{D}$ for various initial conditions, sorted in increasing order for clarity. This clearly shows the fraction of initial conditions for which the optimized $\bar{D}$ is small (see *Appendix 8—table 1*).

## Appendix 9

### Robustness of optimization to small perturbations

We analyse the sensitivity to small perturbations around the optimal point by calculating the Hessian of the KL divergence,

$$H(i,j) = \left. \frac{\partial^2}{\partial X_i \partial X_j} F \right|_{X_{\min}} \tag{38}$$

Here, $X = [\boldsymbol{\mu}, \boldsymbol{R}, \boldsymbol{L}, \sigma]$ denotes the entire set of optimization variables (note that the entries in $X$ are normalized by their respective range and do not carry physical dimensions). We calculated the eigenvalues, denoted by $\lambda_i$, and eigenvectors, denoted by $\boldsymbol{V_i}$, of the Hessian matrix to identify the stiff and sloppy directions (*Gutenkunst et al., 2007*; *Machta et al., 2013*) in the optimization space. The eigenvectors of the Hessian matrix can be grouped in $\boldsymbol{R}, \boldsymbol{L}, \boldsymbol{\mu}$, and $\sigma$ directions by looking for the most dominant component in the eigenvector. We find that most of the eigenvectors have significant entries along the direction of only one of the optimization variables $\boldsymbol{\mu}, \boldsymbol{R}, \boldsymbol{L}, \sigma$ , for example, in *Figure 6a*, the eigenvectors 21–36 have significant entries only in the $\boldsymbol{L}$ directions. There is, however, a small number of eigenvectors that have entries over more than one optimization direction, for example, the eigenvector with $\sigma$ dominant direction has some $\boldsymbol{\mu}$ component as well (*Figure 6a*).

### Stiff and sloppy directions

We find that the eigenvalues of the eigenvectors dominated by $\sigma$ and some $\boldsymbol{\mu}$, $\boldsymbol{L}$ directions are orders of magnitude higher than for those dominated by the $\boldsymbol{R}$ directions (see *Figure 6b*). This suggests that $\bar{D}$ has a valley-like structure around the optimal, with $\boldsymbol{R}$ and some $\boldsymbol{L}$ being the flat or sloppy directions.

The fact that enzyme specificity $\sigma$ and some of the $\boldsymbol{L}$ directions are stiff should not be surprising since the typical width and position of peaks in the synthesized distribution are primarily controlled by $\sigma$ and $\boldsymbol{L}$. We have already shown that $\bar{D}$ is a sharp convex function of $\sigma$ for low values of $(N_E, N_C)$ (see *Figure 3*), which gradually flattens out as we increase $(N_E, N_C)$.

The fact that transport rate $\boldsymbol{\mu}$ is a stiff direction is surprising! The stiffness in $\boldsymbol{\mu}$ is due to the fact that the optimal $\boldsymbol{\mu}$ is always at the lower bound, and with even slight increase in $\boldsymbol{\mu}$, the transport becomes too fast for the reactions to be able to produce the intermediate products. For the $(\boldsymbol{R}, \boldsymbol{L})$-dominated eigenvectors, there are bands of sloppy direction and stiff directions. We define the average stiffness in $\boldsymbol{\mu}, \boldsymbol{R}, \boldsymbol{L}$, and $\sigma$ by a weighted average of eigenvalues, where the weight is given by the strength of the corresponding components of the eigenvector.

$$\langle \lambda \rangle_\mu = \ln \left( \sum_i w_i^{(\mu)} \lambda_i \right), \ \langle \lambda \rangle_R = \ln \left( \sum_i w_i^{(R)} \lambda_i \right), \ \langle \lambda \rangle_L = \ln \left( \sum_i w_i^{(L)} \lambda_i \right), \ \langle \lambda \rangle_\sigma = \ln \left( \sum_i w_i^{(\sigma)} \lambda_i \right)$$

Here, $w_i^{(\mu)} = \sum_{j \in \mu} |V_{i,j}| / \sum_j |V_{i,j}|$, $w_i^{(R)} = \sum_{j \in R} |V_{i,j}| / \sum_j |V_{i,j}|$, $w_i^{(L)} = \sum_{j \in L} |V_{i,j}| / \sum_j |V_{i,j}|$ and $w_i^{(\mu)} = \sum_{j \in \sigma} |V_{i,j}| / \sum_j |V_{i,j}|$.

*Figure 6c* shows $\langle \lambda \rangle_\mu$, $\langle \lambda \rangle_R$, $\langle \lambda \rangle_L$, and $\langle \lambda \rangle_\sigma$ as a function of $N_C$ for fixed $N_E = 4$. The average stiffness in $\boldsymbol{R}$ directions, $\langle \lambda \rangle_R$, is considerably lower than the average stiffness in $\sigma$, $\boldsymbol{\mu}$ and $\boldsymbol{L}$ directions. $\sigma$ is the stiffest direction but the stiffness decreases on increasing the $N_C$. Interestingly, the stiffness along $\boldsymbol{L}$ directions increases on increasing $N_C$.

We now define the total average stiffness $\langle \lambda \rangle = \log(\frac{\sum \lambda_i}{N_C + 2N_E N_C + 1})$, that is, log of the sum of the eigenvalues divided by the dimension of the optimization problem, in the space of $N_E, N_C$ . We find that the average stiffness is higher for low values of $(N_E, N_C)$ as compared to higher values of $(N_E, N_C)$, with a few exceptions; and eventually, the average stiffness settles to a fixed low value (*Figure 6d*).

## Appendix 10

### Insensitivity of diversity to threshold $c_{th}$

Since the threshold $c_{th}$ used to count the number of glycan species in the definition of *diversity* is arbitrary (see section 'Strategies to achieve high glycan diversity'), we here show that the qualitative results we obtain are independent of this choice (***Appendix 10—figure 1***).

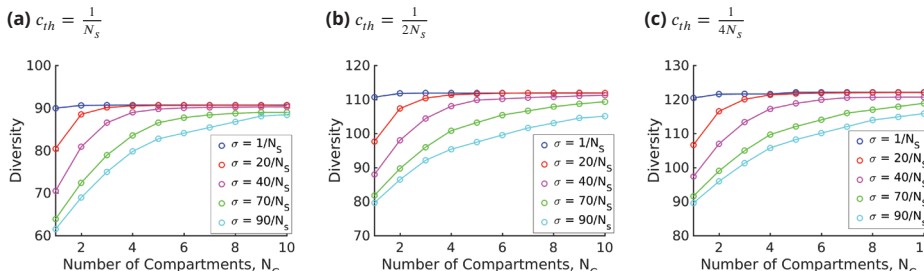

**Appendix 10—figure 1.** Diversity vs. $N_C$ for different values of $\sigma$ keeping $N_E = 1$ fixed, for three different values of the threshold, $c_{th} = \frac{1}{N_s}, \frac{1}{2N_s}, \frac{1}{4N_s}$. Changing the value of the threshold $c_{th}$ only changes the saturation value of the diversity curve.

# Appendix 11

## Table of symbols used in the article (*Appendix 11—table 1*)

**Appendix 11—table 1.** Table of symbols and their definitions.

| Symbol | Definition |
|---|---|
| $c_k^{(j)}$ | Concentration of $k$th glycan in $j$th compartment |
| $\mu^{(j)}$ | Transport rate from $j$th to $j+1$ compartment |
| $\sigma$ | Specificity of the enzymes |
| $\ell_\alpha^{(j)}$ | Ideal substrate length for αth enzyme in $j$th compartment |
| $M(j, k, \alpha)$ | Enzyme parameter related to the Michaelis constant $K_M$ |
| $V(j, k, \alpha)$ | Enzyme parameter related to $V_{max}$ |
| $R(j, \alpha)$ | Reaction parameter for *Optimization B* |
| $D(\mathbf{c}^* \| \mathbf{c})$ | KL divergence between $c^*$ and $c$ |
| $F(\mathbf{c}^* \| \mathbf{c})$ | *Fidelity*, KL divergence normalized by the entropy of the target $c^*$ |
| $\bar{D}(\sigma, N_E, N_C, \mathbf{c}^*)$ | $min_{\boldsymbol{\mu}, \mathbf{R}, \mathbf{L}} F(\mathbf{c}^* \| \mathbf{c})$ |
| $R_{eff}(j, k)$ | Effective reaction rate of $k$th reaction in $j$th compartment |

