## [Editor Report]

This article contributes to an important and largely unexplored topic in cell biology: the understanding of glycosylation. The authors introduce a mathematical model of glycosylation in the Golgi apparatus and use the model to investigate how the complexity (diversity) and fidelity of the plasma membrane glycan distribution depend on parameters such as the number of Golgi cisternae or enzyme specificity. The article is well written and makes the effort to present a rather complex topic in an accessible way by leaving some of the details in the appendices.

---

## [Decision Letter]

**Decision letter after peer review:**

[Editors’ note: the authors submitted for reconsideration following the decision after peer review. What follows is the decision letter after the first round of review.]

Thank you for submitting your work entitled "Glycan processing in the Golgi – optimal information coding and constraints on cisternal number and enzyme specificity" for consideration by *eLife*. Your article has been reviewed by 3 peer reviewers, and the evaluation has been overseen by a Reviewing Editor and a Senior Editor. The reviewers have opted to remain anonymous.

This manuscript presents a clever approach to study from a theoretical point of view how glycosylation reactions occurring at the Golgi membranes define a cell-specific glycan profile. In particular, by optimizing a metric, the authors find that larger number of enzymes and/or compartments (Golgi cisternae) are required for increasingly complex glycan profiles. In addition, they found that enzymes should not be very specific nor too sloppy. This manuscript touches upon an important topic in cell biology (origin of glycan diversity and link to Golgi complex architecture), which has been relatively unexplored in the past. However, although there are no major concerns about the validity of the model presented in the paper (see separate reviews for more detailed comments), there is a general consensus among the reviewers that there are some key points that this manuscript should resolve before opting for publication at *eLife*. Since these points will most likely require an extensive revision of the manuscript, we decided to proceed with a rejection at this moment. We encourage you, if you wish, to resubmit a stronger new paper in the future; or to opt for submission elsewhere for its timely publication.

The main concerns raised by the reviewers are the following:

1) It is not clear what new biology we learn from the results of this paper (see e.g. Rev. 1's report). It's been suggested that the reason for stacking is to give proteins enough time to undergo proper glycosylation and that cisternae number correlates with the complexity of glycans. Since the paper does not include any new experimental evidence on how the complexity of cellular glycans leads to more cisternae in the Golgi stack, it would be very valuable for a biological audience if the authors could present a list of model predictions that could be experimentally tested.

2) For the results of the model to be more robust and probably to also learn a bit more about cell specific glycobiology, a more systematic study of available glycan profiles would very much improve the story (see in particular, Rev. 2, point 2; and also Rev. 3, point 3).

3) The manuscript is very complex for non theorists, since it is presented as a rather technical manuscript. It is well written and organized for specialists. However, it is not clear whether this manuscript, in the current form, will be of interest for a broad biological audience. Along these lines, the conclusions and discussion might need to be toned down, given the large number of simplifications of the model (which is fine for a theoretical audience, but could be over-assumed by a non-specialist biological audience) (see Rev. 3, point 1).

*Reviewer #1:*

The manuscript by Yadav et al. uses a theoretical model of glycan processing to investigate the constraints placed on the enzymatic modification process in order to achieve the complex final glycan distribution. The authors introduce a multi-compartment, many-species model of chemical modifications and develop an optimization algorithm to minimize the divergence between the resulting glycan distribution and that observed from data on immune cells. The authors find that a minimum number of enzymes and reactions chambers (cisternae) are necessary to recapitulate the complex features of the experimental distribution. They also find that enzymes should have an optimal specificity: they should not be too specific to only one particular substrate, nor should they be too "sloppy" by interacting with all substrates with similar rates. Finally, the authors elucidate control parameters of their model that maximize the diversity of glycans.

I am torn about this study because while the approach is elegant and principled and the results sensible and interpretable, the scope is rather technical and therefore I am skeptical as to whether this will have the broad audience and impact that is expected for *eLife*. The work combines elements from several fields (cell biology, chemical modeling, computational optimization, information theory), which is commendable, well-presented, and appropriate for an interdisciplinary audience. At the same time, however, I found myself wondering what we really learn at the end of the day. That the components must be complex (sufficient number of enzymes and compartments) in order for the result to be complex (many-peaked glycan distribution) seems unavoidable. That the specificity must be moderate also seems expected in order to not be limited by a one-to-one matching between enzyme and glycan product, but also not produce a broad, sloppy distribution. Part of the problem may also be that it is unclear what contribution is made in terms of predictions for new experiments (which is often a necessary criterion for a high-impact theoretical study): apart from using actual data as an optimization target, the study largely recapitulates and rationalizes existing features of the glycan machinery, rather than suggesting perturbations for, or placing quantitative limits on, new experimental investigations. My inclination is therefore to suspect that this work may be more suitable for a more specialized journal, e.g. on cell biology or chemical physics.

*Reviewer #2:*

The manuscript "Glycan processing in the Golgi – optimal information coding and constraints on cisternal number and enzyme specificity" by Yadav et al. presents a model for glycan production by the Golgi cisternae. Glycans serve various cellular functions, such as markers for cell identity, and it is important that a function-specific profile (distribution) of glycans is obtained at the end of the production process. The authors show how to choose the reaction constants in their model to reach a target profile (distribution) of glycans. They discuss how the adequacy between the target and model profiles depend on the physico-chemical features of the reactants.

While the problem studied by the authors is interesting and their analysis reports valuable qualitative results, e.g. on the consequence of enzyme specificity on the profile statistics, I list below various points that I think would require further studies and discussions:

1. The writing of the manuscript is quite technical in some sections, in particular around equation (5) and (6), and involves superfluous notations: what is the added value in introducing V, M, L with respect to the already defined omega, P, …? I would recommend the authors to simplify the writing and really emphasize what is important here. Several definitions or assumptions remain vague and not discussed. For instance, equation (3) define the binding probability between enzyme α and substrate k in terms of "shape" vectors. On line 250 these shape vectors are defined as scalars, in particular the shape of enzyme k is identified with the index itself. Why is this a good choice (apart from reducing the dimension of the parameter space), and how could one test the validity this apparently quite strong hypothesis?

2. The manuscript is mostly theoretical with limited connections with experimental data and validations. It would be nice to have a systematic study of available glycan profiles and comparison of the corresponding inferred model parameters. In addition, it is necessary to validate the inference/optimization procedure on synthetic profiles generated by stochastic simulations of some prescribed network, to check that the correct network parameters are correctly recovered (or some other solutions giving back the same profile, see point 4 below).

3. While the KL divergence (7) is a reasonable measure of dissimilarity between two distributions the relevance of the ratio (9) on page 10 is not at all clear to me. More precisely, I do not see why the entropy of the target distribution is the relevant quantity here. Rather the KL divergence between the model and target distribution could be compared to the mean value of the KL divergence between any two glycan profiles resulting from the noisy production steps.

4. Since the function of the minimize over, see eq. (8) is non convex, there may exist many local minima. The authors report in Appendix 8 they perform grid-like search. How close are those minima in terms of the objective function? This question is important as the model studied here seems to be overparametrized, in particular when N_c is large. It may also be relevant in terms of evolution. More precisely, how were some glycan profiles and the corresponding biochemical networks selected in the course of evolution? On top of selection pressure to "orthogonalize" the profiles is it possible that some distributions are (approximately) realized by more networks than others and were therefore entropically favored?

*Reviewer #3:*

The paper by Yadav et al. contributes to an important and largely unexplored topic in Life Sciences, namely, to the understanding of glycosylation. The authors introduce a mathematical model of glycosylation in the Golgi apparatus and use the model to investigate how the complexity (diversity) and fidelity of the plasma membrane glycan distribution depends on parameters such as the number of Golgi cisternae or enzyme specificity. The paper is well written and makes the effort to present a rather complex topic in an accessible way by leaving some of the details in the appendices.

After defining the quantitative measures for complexity and fidelity of glycan distributions, the paper builds a general model of glycosylation in the Golgi by assuming Michaelis-Menten type reactions and by taking into account parameters such as the number, specificity and distribution of enzymes across cisternae. However, the parameter space becomes so large that, before solving it numerically, the authors need to reduce it by making a number of simplifications (they linearize the distortion energy; assign one enzyme specificity σ to all reactions …). The authors then ask the question how different parameters influence the outcome and what are the optimal parameters for a given target glycan distribution. Finally, the authors explore the strategies to achieve high glycan diversity.

My comments are listed below:

1. The authors should be more careful (modest?) with the generalization of their results and conclusions. The results are obtained by a very simplified model and it is not obvious that they can be generalized.

For example, in the Abstract they state: "We find that to synthesize complex distributions, such as those observed in real cells, one needs to have multiple cisternae and precise enzyme partitioning in the Golgi." However, this is only true within the simplified model. Maybe, complex distributions could be obtained even with a small number of cisternae if one allowed biochemistry that is more complex. The statement above would be more accurate if it was turned around, e.g., "we show that multiple cisternae and precise enzyme partitioning in the Golgi can lead to complex glycan distributions."

The readers of a mathematical journal would very well understand that the results are not general but should be interpreted within the model proposed. On the other hand, the readers of *eLife* could be misled by statements which are too general. This is even more true for this paper, which starts with a rather general and complex model and then simplifies it on the go. While the authors do summarize the simplifications in the Discussion, an inexperienced reader could still miss them and take the results too literally. I would suggest that the authors amend the whole paper according to this comment to avoid possible misinterpretations.

2. An interesting result of the modeling is that the role of parameters NE and NC can be rather symmetrical – the fidelity of glycan distribution can be improved in a similar way by increasing either of these parameters (Figure 4). Why is that? Is this a consequence of a symmetrical role of these parameters in the reduced (simplified) parameter space, or can this symmetry be generalized and can have a biological merit?

3. The paper mentions examples of cells that have a complex Golgi and a complex glycan distribution. Is the opposite also true, e.g., that cells with a simple Golgi do not have complex glycan distributions? Or can a complex Golgi sometimes result in a low glycan complexity?

4. One wonders how robust the glycosylation process is according to the model proposed? Figure 5b neatly shows that large enzyme repartitioning can lead to a completely different glycan distribution. But how do small variations of the optimal reaction distribution affect the outcome? Would it be possible to quantify the robustness and analyze it systematically (not just show a few random examples)? How is robustness affected by the number of cisternae and enzymes? What are biological implications of this result?

5. Figure 5b shows that a complex Golgi can also lead to a simple glycan distribution. This implies that having a large NC and NE is not sufficient for a complex glycan distribution. Could the authors discuss this? What are additional requirements that are needed for a complex glycan distribution?

6. What is "likelihood," which is used in some of the graphs? Is it related to the minimum achievable Kullback-Leibler (KL) divergence according to Kullback-Leibler metric defined in Eq. 7 (and used in Figures 3, 4 and 5)? If so, the authors could use only one measure of likelihood/divergence in all the graphs.

7. Why does the mouse glycan distribution have a lower "likelihood" than the human one, even with an increasing number of GMM (Figure 1b)? Is this a coincidence or can this be generalized?

8. The paper would be easier to read if all the parameters were listed in one place, and if one letter was not used to denote different things (e.g., "k" denotes complexity, but it is also used as the index for glycosylation reactions).

9. Is the final distribution of glycans on PM presented on the right in Figure 2 conceptually the same as the glycan distributions presented in Figure 1? If so, use the same notation (and axes labels) everywhere.

---

## [Author Response]

[Editors’ note: the authors resubmitted a revised version of the paper for consideration. What follows is the authors’ response to the first round of review.]

Reviewer #1:The manuscript by Yadav et al. uses a theoretical model of glycan processing to investigate the constraints placed on the enzymatic modification process in order to achieve the complex final glycan distribution. The authors introduce a multi-compartment, many-species model of chemical modifications and develop an optimization algorithm to minimize the divergence between the resulting glycan distribution and that observed from data on immune cells. The authors find that a minimum number of enzymes and reactions chambers (cisternae) are necessary to recapitulate the complex features of the experimental distribution. They also find that enzymes should have an optimal specificity: they should not be too specific to only one particular substrate, nor should they be too "sloppy" by interacting with all substrates with similar rates. Finally, the authors elucidate control parameters of their model that maximize the diversity of glycans.I am torn about this study because while the approach is elegant and principled and the results sensible and interpretable, the scope is rather technical and therefore I am skeptical as to whether this will have the broad audience and impact that is expected for eLife. The work combines elements from several fields (cell biology, chemical modeling, computational optimization, information theory), which is commendable, well-presented, and appropriate for an interdisciplinary audience. At the same time, however, I found myself wondering what we really learn at the end of the day. That the components must be complex (sufficient number of enzymes and compartments) in order for the result to be complex (many-peaked glycan distribution) seems unavoidable. That the specificity must be moderate also seems expected in order to not be limited by a one-to-one matching between enzyme and glycan product, but also not produce a broad, sloppy distribution. Part of the problem may also be that it is unclear what contribution is made in terms of predictions for new experiments (which is often a necessary criterion for a high-impact theoretical study): apart from using actual data as an optimization target, the study largely recapitulates and rationalizes existing features of the glycan machinery, rather than suggesting perturbations for, or placing quantitative limits on, new experimental investigations. My inclination is therefore to suspect that this work may be more suitable for a more specialized journal, e.g. on cell biology or chemical physics.

We are happy that the referee finds the paper “elegant”, “commendable” and “appropriate for an interdisciplinary audience”. Despite this the referee does not feel that the manuscript provides new predictions or suggestions for new experiments and simply “rationalizes existing features of the glycan machinery”.

While it is true that our study provides rationalisations for existing (but not well appreciated) features of the cellular glycan machinery, it does so from a fresh perspective, namely from a distinct information theoretic point of view. This should not be considered as a problem, since one of the roles of theory is to view disparate observations through the lens of a general framework. Our abstract information theoretic considerations are a novel conceptual way of approaching the biochemical synthesis of information carrying molecules, and make quantitative, the purely verbal descriptions in, 2. We hope such a perspective would prove useful in the future.

In addition, our theoretical study also makes testable predictions, provides suggestions for future experiments and opens up a scope for new enquiry. Our rewriting of the manuscript emphasises this aspect – we do this by (i) including and analysing more data from different species, that together highlight the relation between organismal/glycan complexity and the complexity of the Golgi machinery; and (ii) highlighting the predictions and suggestions that follow from this study (this has involved some more calculations). While the abstraction and generality of the description helps in bringing together several aspects and highlighting new questions, it can at best make qualitative predictions. Our task therefore has been to to get at a qualitative understanding using quantitative methods and thereby to arrive at general principles. This approach has been highly successful in several branches of physics, a rather famous example is the Landau theory of phase transitions, and has been passionately, and we think correctly, advocated for Biology in [3, 4].

We now highlight the new conceptual contributions, predictions and offshoots/implications of our theoretical study and suggestions for future experiments -

1. Since an important function of the glycan spectrum is cell type/niche identification, it seems natural to relate *glycan complexity* to organismal complexity taken to be associated with the number of cell types in the organism [5, 6]. Here, we provide a *measure of the complexity* of the glycan distribution of a given cell type using MSMS data. Using this we have now analysed the MSMS data from Hydra, Planaria, Mammalian cells.

2. We have constructed a generic and calculable model of *chemical synthesis* of the glycan spectrum (molecular information) consistent with the phenomenology of enzymatic reactions and cellular compartmentalization. This represents a calculable model, as opposed to a black-box model, of the Golgi machinery.

3. Constructing a high fidelity representation of a *complex target distribution*, such as those observed in real cells, requires a *complex Golgi machinery* with multiple cisternae, precise enzyme partitioning and control on enzyme specificity. This definition of fidelity of the glycan code, allows us to provide a quantitative argument for the evolutionary requirement of multiple-compartments. While it is possible to produce complex glycan distributions in one compartment using a large number of enzymes, such a design would inevitably require a more elaborate genetic cost.

4. One of the predictions of the model is that we should see a qualitative trend of increasing Golgi compartments on increasing the glycan complexity, keeping everything else fixed. This can be empirically verified by looking at the glycan profiles, ultrastructure of Golgi and the number of glycosylation enzymes across many species.

5. The model solution is degenerate, in the sense that there many equally good global minima. These degeneracies are both continuous and discrete. The continuous degeneracies corresponds to regions in the reaction rate – transport rate space, moving along which will not change the concentration profile, thus ensuring *robustness* to internal noise. This suggests that the distribution is robust to slight cell-to-cell variations in these kinetic parameters.

6. Our model implies that close to a local minima the inter-cisternal transport rate ***µ*** and the specificity of the enzymes *σ* are stiff directions, i.e. the cell should exercise tighter control on ***µ*** and *σ* as compared to the reaction rates. The reaction rates close to the local minima are sloppy directions, moving along these directions does not change the glycan profile much.

7. For fixed number of enzymes and cisternae, there is an optimal level of specificity of enzymes that achieves the complex target distribution with high fidelity. Keeping the number of enzymes fixed, having low specificity or sloppy enzymes and larger cisternal number could give rise to a diverse repertoire of functional glycans, a strategy used in organisms such as plants and algae. Sloppy or promiscuous enzymes bring in the potential for *evolvability* [7], sloppiness allows the system to be stable to random mutations in proteins or variations in the target distribution.

8. Taken together, our quantitative analysis of the tradeoffs has profound implications for nonequilibrium self assembly of the Golgi cisternae, suggesting that the nonequilibrium control of cisternal number must involve a coupling of nonequilibrium self assembly of cisternae with enzymatic chemical reaction kinetics [8].

Reviewer #2:The manuscript "Glycan processing in the Golgi – optimal information coding and constraints on cisternal number and enzyme specificity" by Yadav et al. presents a model for glycan production by the Golgi cisternae. Glycans serve various cellular functions, such as markers for cell identity, and it is important that a function-specific profile (distribution) of glycans is obtained at the end of the production process. The authors show how to choose the reaction constants in their model to reach a target profile (distribution) of glycans. They discuss how the adequacy between the target and model profiles depend on the physico-chemical features of the reactants.While the problem studied by the authors is interesting and their analysis reports valuable qualitative results, e.g. on the consequence of enzyme specificity on the profile statistics, I list below various points that I think would require further studies and discussions:1. The writing of the manuscript is quite technical in some sections, in particular around equation (5) and (6), and involves superfluous notations: what is the added value in introducing V, M, L with respect to the already defined omega, P, …? I would recommend the authors to simplify the writing and really emphasize what is important here.

We thank the referee for this suggestion. *V* and *M* are physically measurable macroscopic quantities (vectors) related to the Michaelis constant and maximum velocity *V_max_* of the enzymes, for which we place bounds that we take from literature (see Appendix 2). These are defined in terms of the microscopic rates *ω_f_,ω_b_* and *ω_c_*, see equation 4 of the manuscript. However to make the manuscript clearer we have now relegated the expression for the concentrations in terms of *M*,*V* and *L* (equations 5 and 6 of the manuscript) to Appendix 3. We have also simplified the writing as recommended, highlighting essential features.

Several definitions or assumptions remain vague and not discussed. For instance, equation (3) defines the binding probability between enzyme α and substrate k in terms of "shape" vectors. On line 250 these shape vectors are defined as scalars, in particular the shape of enzyme k is identified with the index itself. Why is this a good choice (apart from reducing the dimension of the parameter space), and how could one test the validity this apparently quite strong hypothesis?

We now state all the assumptions that we are making at the beginning of the Results section. In general, the shape of an object (enzyme) can be described by a shape tensor akin to a moment of inertia tensor (or a fabric tensor that expands the shape in spherical harmonics). Restricting the shape deformations to a scalar is tantamount to a single mode approximation. This is essentially the approximation used when a single reaction coordinate is taken to represent a multidimensional reaction landscape.

In a chemical reaction between macromolecules, the reaction centres are often buried in the molecule and not easily accessible, unless reactants have the appropriate shape (lock-key) or can be induced to have the appropriate shape (induced fit). Thus, reactions rates depend on the shape and its deformability, which may be modelled by a general shape tensor (we call this distortion). A simple model for distortion is a scalar model, as used for instance in [9]. We have chosen to parametrize the optimal length of the enzyme by a integer multiple of a basic unit. This choice merely simplifies the analysis. Normally in the analysis of chemical kinetics, this would be subsumed in the rates, since the experimentally measured rates are a function of the shape distortion, pH etc. Since in our work, we are interested in analysing the tradeoffs involving enzymatic parameters, we need to provide a “microscopic” model of enzyme reactions that allows us to proceed with our optimization analysis in a tractable manner.

2. The manuscript is mostly theoretical with limited connections with experimental data and validations. It would be nice to have a systematic study of available glycan profiles and comparison of the corresponding inferred model parameters.

What the referee is asking for is precisely the goal of glycan engineering (see Discussion in the revised manuscript). This is a mature and serious field of study, and has contributed a lot to understanding the chemical profile of glycans in health and disease. There are a number of sequential enzymatic reaction models [10, 11, 12] that predict the N-glycan distribution based on the activities and levels of processing enzymes distributed in the Golgi-cisternae of mammalian cells, and make comparisons with N-glycan mass spectrum data. Models such as the KB2005 model [10, 11, 12] are extremely elaborate (with 22800 chemical reactions) and require very detailed chemical input parameters, which are either determined experimentally or obtained by fitting to the mass spectrometry data. These models have an important practical role to play, that of being able to predict the impact of the various *chemical parameters* on glycan distribution.

In contrast, here we pose a different question: Given an evolutionary drive for a specific cell type within a specific tissue microenvironment to maintain a target glycan distribution of fixed complexity, what are the tradeoffs in the chemical and physical parameters of glycan biosynthesis that realise this target.

This engenders a different strategy of analysis as advocated here, and allows us to get at a qualitative understanding using quantitative methods, as elaborated in the Response to Referee 1.

In addition, it is necessary to validate the inference/optimization procedure on synthetic profiles generated by stochastic simulations of some prescribed network, to check that the correct network parameters are correctly recovered (or some other solutions giving back the same profile, see point 4 below).

We agree with the referee and thank the referee for this suggestion. We have now added a new section in the manuscript (Please see Appendix 8 of the revised manuscript).

3. While the KL divergence (7) is a reasonable measure of dissimilarity between two distributions the relevance of the ratio (9) on page 10 is not at all clear to me. More precisely, I do not see why the entropy of the target distribution is the relevant quantity here. Rather the KL divergence between the model and target distribution could be compared to the mean value of the KL divergence between any two glycan profiles resulting from the noisy production steps.

The entropy of target distribution is a measure of the total statistical information in the distribution. Normalizing with respect to the entropy allows us to measure the information gained about the target distribution by approximating it with the empirical distribution, as the fraction of the total statistical information in the target distribution. This is a reasonable measure for comparing the effectiveness of the model for different target distributions. We have now made this point more explicit in the revised manuscript (Please see Equation 5 in section "Optimization Problem").

4. Since the function of the minimize over, see eq. (8) is non convex, there may exist many local minima. The authors report in Appendix 8 they perform grid-like search. How close are those minima in terms of the objective function? This question is important as the model studied here seems to be overparametrized, in particular when N_c is large. It may also be relevant in terms of evolution. More precisely, how were some glycan profiles and the corresponding biochemical networks selected in the course of evolution? On top of selection pressure to "orthogonalize" the profiles is it possible that some distributions are (approximately) realized by more networks than others and were therefore entropically favored?

The issue of parameter degeneracy and possible evolutionary implications is very interesting, and we thank the referee for bringing this up. We have conducted a detailed study, which we now report in Appendix 8 subsection “Degeneracy in the model”.

Reviewer #3:The paper by Yadav et al. contributes to an important and largely unexplored topic in Life Sciences, namely, to the understanding of glycosylation. The authors introduce a mathematical model of glycosylation in the Golgi apparatus and use the model to investigate how the complexity (diversity) and fidelity of the plasma membrane glycan distribution depends on parameters such as the number of Golgi cisternae or enzyme specificity. The paper is well written and makes the effort to present a rather complex topic in an accessible way by leaving some of the details in the appendices.After defining the quantitative measures for complexity and fidelity of glycan distributions, the paper builds a general model of glycosylation in the Golgi by assuming Michaelis-Menten type reactions and by taking into account parameters such as the number, specificity and distribution of enzymes across cisternae. However, the parameter space becomes so large that, before solving it numerically, the authors need to reduce it by making a number of simplifications (they linearize the distortion energy; assign one enzyme specificity σ to all reactions …). The authors then ask the question how different parameters influence the outcome and what are the optimal parameters for a given target glycan distribution. Finally, the authors explore the strategies to achieve high glycan diversity.My comments are listed below:1. The authors should be more careful (modest?) with the generalization of their results and conclusions. The results are obtained by a very simplified model and it is not obvious that they can be generalized.For example, in the Abstract they state: "We find that to synthesize complex distributions, such as those observed in real cells, one needs to have multiple cisternae and precise enzyme partitioning in the Golgi." However, this is only true within the simplified model. Maybe, complex distributions could be obtained even with a small number of cisternae if one allowed biochemistry that is more complex. The statement above would be more accurate if it was turned around, e.g., "we show that multiple cisternae and precise enzyme partitioning in the Golgi can lead to complex glycan distributions."The readers of a mathematical journal would very well understand that the results are not general but should be interpreted within the model proposed. On the other hand, the readers of eLife could be misled by statements which are too general. This is even more true for this paper, which starts with a rather general and complex model and then simplifies it on the go. While the authors do summarize the simplifications in the Discussion, an inexperienced reader could still miss them and take the results too literally. I would suggest that the authors amend the whole paper according to this comment to avoid possible misinterpretations.

We apologise for not being careful in stating our results and presenting the necessary caveats. We believe we have now worked on it to a considerable extent in the revised submission.

We find that even within the framework of our simple model it is possible to generate complex distributions in a small number of compartments by having a complex biochemistry (large number of processing enzymes). However, this complex biochemistry invokes an elaborate genetic cost on the cell. Therefore, within our framework the number of compartments required are both a function of the complexity of glycan distribution to be synthesized and the number of enzymes. This is a crucial insight and we have edited the manuscript to make this point and the underlying assumptions explicitly.

2. An interesting result of the modelling is that the role of parameters NE and NC can be rather symmetrical – the fidelity of glycan distribution can be improved in a similar way by increasing either of these parameters (Figure 4). Why is that? Is this a consequence of a symmetrical role of these parameters in the reduced (simplified) parameter space, or can this symmetry be generalized and can have a biological merit?

We thank the referee for bringing this up. This is an interesting observation. However, we should hasten to add that this apparent symmetry is a consequence of our simplified chemical and transport model, as we have verified by making numerous extensions of the model. We now discuss these issues in detail in the section "Tradeoffs between number of enzymes, number of cisternae and enzyme specificity to achieve given complexity" of the revised manuscipt.

3. The paper mentions examples of cells that have a complex Golgi and a complex glycan distribution. Is the opposite also true, e.g., that cells with a simple Golgi do not have complex glycan distributions? Or can a complex Golgi sometimes result in a low glycan complexity?

Since the complexity of the glycan distribution depends on *both* the complexity of the Golgi and the complexity of the chemical processing, this is a very difficult question to answer. We do feel that a systematic analysis of the enzyme chemistry, Golgi morphology and transport would shed more light on this issue, and indeed this is one of the suggestions that emerge from our analysis. For this paper, we have taken the glycan MS data of of a few cell types – planaria, hydra, mouse and human blood cells.

4. One wonders how robust the glycosylation process is according to the model proposed? Figure 5b neatly shows that large enzyme repartitioning can lead to a completely different glycan distribution. But how do small variations of the optimal reaction distribution affect the outcome? Would it be possible to quantify the robustness and analyze it systematically (not just show a few random examples)? How is robustness affected by the number of cisternae and enzymes? What are biological implications of this result?

Again, we thank the referee for bringing up this very interesting issue. We now provide a detailed analysis of the sensitivity to small perturbations around the optimal solution by calculating the Hessian of the KL divergence in the new section "Geometry of the Fidelity landscape" of the revised manuscript.

7. Why does the mouse glycan distribution have a lower "likelihood" than the human one, even with an increasing number of GMM (Figure 1b)? Is this a coincidence or can this be generalized?

Apart from updating the GMM procedure, we have also included data from two different organisms, *Hydra* and *Planaria*, in our study. Comparisons across these species tell a story when viewed from our framework.

Figure 5 compares the complexity of glycan profiles of Hydra, Planaria and Humans. Number of cell types in Hydra, Planaria and Humans are around 41 [15], 44 [16] and 103 [17] respectively, based on transcriptome analysis (these are lower bounds based on the main cell types, and especially for Planaria and Hydra, are subject to constant revision). These studies cluster thousands of cells from various tissues in the organism on the basis of statistical similarity of mRNA expression in the cell, with each cluster representing a cell type. This method is quite good for estimating the cell types in simpler organisms or single tissues but undercounts the number of cell types for humans due to limited cell number of each cell type in the sample. Based on morphology the number of cell types in humans is estimated to be around 200 [18].

Our analysis of the MSMS data of these organisms suggest that organism with fewer cell types have less complex glycan distribution.

It would be interesting to determine systematic correlations between complexity of Golgi, number of glycosylation enzymes and the complexity of the glycan distribution, in organisms such as algae (which typically possess many cisternae).

We have included this figure and a related discussion in Appendix 1 of the revised manuscript.

[1] Gabius, Hans-Joachim. ”The sugar code: Why glycans are so important.” Biosystems 164 (2018): 102111.

[2] Bard, Frederic, and Joanne Chia. ”Cracking the glycome encoder: signaling, trafficking, and glycosylation.” Trends in cell biology 26.5 (2016): 379-388.

[3] Sengupta, Anirvan M. Modeling biomolecular networks: an introduction to systems biology. Oxford University Press, 2008.

[4] W. Bialek, W., ”Biophysics: Searching for Principles”, (Princeton University Press, Princeton, 2002).

[5] Carroll, Sean B. ”Chance and necessity: the evolution of morphological complexity and diversity.” Nature 409.6823 (2001): 1102-1109.

[6] Bonner, John Tyler. ”The origins of multicellularity.” Integrative Biology: Issues, News, and Reviews: Published in Association with The Society for Integrative and Comparative Biology 1.1 (1998): 27-36.

[7] Kirschner, M.W. and Gerhart, J.C., ”The plausibility of life”, (Yale University Press, New Haven, 2005).

[8] Glick, B.S. and Malhotra, V., ”The curious status of the Golgi apparatus”, Cell 95: 883-889 (1998).

[9] Savir Y, Tlusty T. "Conformational proofreading: the impact of conformational changes on the specificity of molecular recognition". PloS one. 2007; 2(5):e468

[10] Umana, Pablo, and James E. Bailey. "A mathematical model of N linked glycoform biosynthesis." Biotechnology and bioengineering 55.6 (1997): 890-908.

[11] Krambeck, Frederick J., and Michael J. Betenbaugh. "A mathematical model of N linked glycosylation." Biotechnology and Bioengineering 92.6 (2005) 711-728.

[12] Krambeck, Frederick J., et al. "A mathematical model to derive N-glycan structures and cellular enzyme activities from mass spectrometric data." Glycobiology 19.11 (2009): 1163-1175.

[13] Gutenkunst, Ryan N., et al. "Universally sloppy parameter sensitivities in systems biology models." PLoS computational biology 3.10 (2007): e189.

[14] Machta, Benjamin B., et al. "Parameter space compression underlies emergent theories and predictive models." Science 342.6158 (2013): 604-607.

[15] Siebert, Stefan, et al. "Stem cell differentiation trajectories in Hydra resolved at single-cell resolution." Science 365.6451 (2019).

[16] C.T. Fincher et al, Cell type transcriptome atlas for the planarian Schmidtea mediterranea, Science. 2018 May 25;360(6391).

[17] Han, Xiaoping, et al. "Construction of a human cell landscape at single-cell level." Nature 581.7808 (2020): 303-309.

[18] Junqueira, Carneiro, Kelley, Basic Histology, 7th edition 1992 Appleton and Lange, p.66